# A humanized anaplastic lymphoma kinase (ALK)-directed antibody-drug conjugate with pyrrolobenzodiazepine payload demonstrates efficacy in ALK-expressing cancers

The *Anaplastic Lymphoma Kinase* (*ALK*) gene is a receptor tyrosine kinase (RTK) with expression restricted to the developing nervous system. Most neuroblastomas express native ALK protein on the cell surface and ALK is uniformly overexpressed in fusion-positive rhabdomyosarcoma and in subsets of metastatic colorectal carcinoma, melanoma, ovarian carcinoma, and breast carcinoma. Here, we first confirm that *ALK* RNA, protein, and tumor cell surface expression is elevated in multiple pediatric and adult malignancies with minimal expression in childhood normal tissues. We then demonstrate that a humanized ALK-directed antibody conjugated to pyrrolobenzodiazepine (CDX0239-PBD) is internalized in ALK-expressing neuroblastoma cell lines with cell surface expression-dependent cytotoxicity. Finally, we show that CDX0239-PBD exhibits potent antitumor efficacy including maintained complete responses in ALK-expressing patient and cell line-derived neuroblastoma, fusion-positive rhabdomyosarcoma, and colorectal carcinoma xenograft models. These data support the clinical development of a first-in-class ALK-directed antibody-drug conjugate (ADC) for multiple pediatric and adult ALK-expressing malignancies.

The *Anaplastic Lymphoma Kinase* (*ALK*) gene is a receptor tyrosine kinase (RTK) with physiologic expression restricted to the developing central and peripheral nervous system[1]. ALK is aberrantly expressed in cancer predominantly through gain-of-function alterations such as mutations or chromosomal translocations, and through high-level amplification in diverse childhood and adult cancers[2-12]. We and others have identified activating point mutations within the tyrosine kinase domain (TKD) as the major cause of hereditary neuroblastoma and as the most common somatic single-nucleotide mutations in the sporadic forms of the disease[2,3,13,14]. Neuroblastoma is an embryonal tumor derived from the neural crest that arises during fetal or early post-natal life from the misappropriation of normal neurodevelopmental pathways[15]. Over half of patients have aggressive high-risk disease with less than 50% survival probability despite intensive chemotherapy, surgery, radiation, and immunotherapy[16,17]. Most survivors of high-risk neuroblastomas are left with long-term and life-threatening morbidities, including a high incidence of secondary malignancies[18,19], demonstrating a clear need for targeted precision therapy to enhance treatment efficacy and reduce off-target toxicities.

Recent ultradeep sequencing of diagnostic tumors from patients with high-risk neuroblastoma demonstrates that oncogenic aberrations are more prevalent than previously recognized, with 25%

e-mail: mosse@chop.edu

harboring an activating *ALK* mutation or gene amplification[20]. These patients have inferior outcomes compared to patients without an *ALK* aberration[20]. Consistent with prior findings, MYCN amplification is more frequent with *ALK*-aberrant tumors, particularly those harboring *ALK* F1174L mutations[21,22]. Previous studies have demonstrated a higher frequency of mutations in *ALK* and RAS-MAPK pathways at relapse, suggesting an enrichment of selected subclonal mutations as a mechanism of therapy resistance[23,24]. Mutant *ALK* has been therapeutically targeted in neuroblastoma with small molecular enzymatic inhibitors that have been implemented from the original discoveries of germline and somatic *ALK* mutations[2,3,13,14], to preclinical proof-of-concept[25,26], through early and late-phase clinical implementation[27,28] (NCT03126916). Notably, native ALK protein is expressed on the cell surface of the vast majority of neuroblastoma tumors[29–31] and as an oncofetal protein, it is not expressed in somatic post-natal tissues, thereby nominating ALK as an ideal immunotherapy target. Furthermore, ALK is expressed on the cell surface of rhabdomyosarcomas harboring a PAX3 or PAX7::FOXO1 fusion (fusion-positive rhabdomyosarcoma), the most common soft tissue sarcoma of childhood and adolescence[4,5]. ALK is also expressed in subsets of metastatic colorectal carcinoma[6], malignant melanoma[7,11], ovarian carcinoma[8–11], and breast carcinoma[8,12], suggesting a broad application for ALK-targeting therapies in multiple childhood and adult solid malignancies.

Therapies against cell-surface targets have emerged as a compelling approach for solid tumors. Antibody-drug conjugates (ADCs) are an innovative class of cancer therapeutics that combine the specificity of monoclonal antibodies with the potency of cytotoxic drugs[32]. Our previously published ALK-directed ADC demonstrated proof-of-concept in neuroblastoma but was limited in clinical translation due to a lack of potent and sustained efficacy with a thienoindole (TEI) payload[29] at doses higher than clinically relevant ADC dosing[33]. Additionally, this ADC utilized a chimeric ALK-directed antibody, which carries an immunogenicity risk compared to humanized antibodies due to the formation of human anti-mouse antibodies that leads to rapid clearance, poor tumor penetration, and hypersensitivity reactions[34]. We remain steadfast in our efforts to exploit ALK as a tumor-restricted antigen by developing an ADC with a humanized ALK-directed antibody with cross-species reactivity to decrease immunogenicity risk, and with conjugation to a cytotoxic payload with maximal potency and a physically stable linker. We hypothesize that an ADC

approach can address the majority of patients with neuroblastoma, in the presence or absence of activating ALK alterations, since most tumors express cell surface native ALK protein[29,30]. We also postulate that ALK functions as a tractable tumor-restricted antigen in other malignancies in addition to neuroblastoma.

Here, we validate ALK as a compelling immunotherapy target in pediatric and adult solid malignancies. This is followed by extensive in vitro and in vivo validation of a humanized ALK-directed ADC, conjugated to the sequence-selective DNA minor-groove binding cross-linking agent pyrrolobenzodiazepine dimers[35] (CDX0239-PBD), in ALK-expressing neuroblastoma, fusion-positive rhabdomyosarcoma and colorectal carcinoma supporting clinical development and a broad therapeutic application of an ALK-directed ADC.

## Results
### ALK is a differentially expressed lineage-restricted antigen in multiple pediatric and adult cancers

To identify additional malignancies beyond neuroblastoma with the potential for application of ALK-directed immunotherapeutic approaches, we first analyzed publicly available RNA sequencing data from the Tumor Compendium v11 of UCSC Treehouse Childhood Cancer initiative and from The Genotype-Tissue Expression Project v8 for neoplastic and normal tissues, respectively. Determination of *ALK* RNA expression was assessed using transcripts per million (TPM) and *ALK* expression outliers included neoplastic histologies within the top 95th percentile (median TPM greater than or equal to 6.56 TPM) or more than 1 sample in the top 99th percentile (TPM greater than or equal to 31.04). Median *ALK* RNA expression outliers was highest in descending order of neuroblastoma (21.64, $n = 201$), fusion-positive rhabdomyosarcoma (17.69, $n = 49$), dysembryoplastic neuroepithelial tumor (8.13, $n = 14$), Ewing sarcoma (6.75, $n = 85$), ganglioglioma (2.62, $n = 47$), glioblastoma multiforme (2.47, $n = 200$), cutaneous melanoma (2.27, $n = 469$), diffuse midline glioma (2.25, $n = 82$), and high-grade glioma (0.72, $n = 149$) with minimal or absent *ALK* RNA expression in other neoplastic histologies (Supplemental Table 1). Our database analysis included breast and ovarian carcinomas and although they were not found to be outliers of *ALK* RNA expression, previous studies have demonstrated native *ALK* RNA expression in subsets of these malignancies[9,11]. *ALK* RNA TPM in all normal tissues ($n = 17,382$ across 31 tissues) were less than neuroblastoma and was generally absent

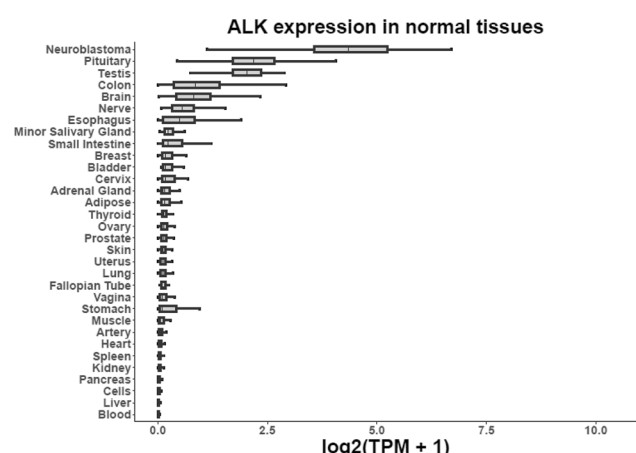

**A**

**Outliers in ALK expression**

**B**

**ALK expression in normal tissues**

**Fig. 1** | ***ALK* RNA is expressed in multiple neoplastic histologies and has restricted expression in normal histologies. A** Neoplastic histologies considered outliers of *ALK* RNA expression defined as median at or greater than the 95th percentile of TPM or one sample at or greater than the 99th percentile of TPM in descending order. **B** Normal histology median *ALK* RNA expression compared to neuroblastoma. Data plotted as median log 2 transformed transcript per million

(TPM) + 1 to account for tissues with a TPM of 0. Box plot center lines define median *ALK* RNA expression, box limits define the third quartile, and whiskers define the minimum and maximum values. Sample size of biologic tumor replicates, median TPM, mean TPM, and standard deviation of TPM are provided in Supplemental Tables 1–2.

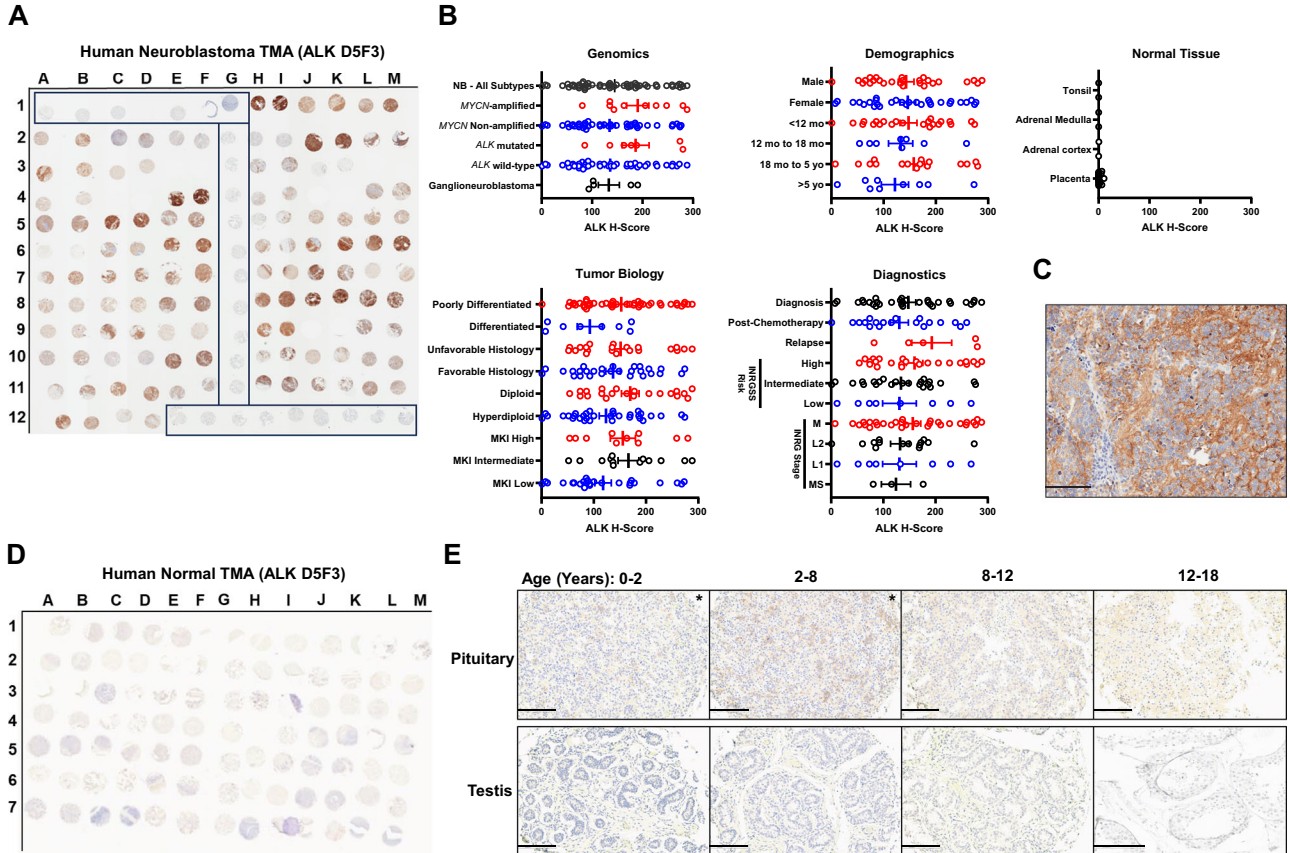

**Fig. 2 | ALK is expressed in multiple subsets of neuroblastoma and has restricted expression in normal tissue. A** Immunohistochemical (IHC) staining for ALK in a human neuroblastoma tissue microarray (TMA) with boxes indicating normal tissue samples. **B** ALK H-scores in all neuroblastoma (NB) tumors, ganglioneuroblastoma tumors, neuroblastoma tumors stratified by genomic, tumor biology, demographic, and diagnostic characteristics, and normal tissues from a human neuroblastoma TMA. The average ALK H-score for each tumor or tissue is plotted individually with bars representing group ALK H-score mean ± standard error of the mean. Average H-scores were compared using a two-away ANOVA analysis with Tukey's multiple comparison test. Sample size of biologic tumor

replicates, datasets, and exact *p*-values for are located within the Source Data file. Source data are provided as a Source Data file. **C** IHC staining for ALK in a representative diagnostic neuroblastoma core with an H-score of 142.33 from the neuroblastoma TMA at 20x magnification (scale bar represents 100 µm, ALK staining for this tumor was completed on two individual tumor cores with one tumor core represented). **D** IHC staining for ALK in a pediatric normal TMA **E** IHC staining for ALK in pediatric pituitary and testis with samples categorized by age group in years (*indicates equivocal staining in a subset of cells, otherwise samples negative for ALK staining; scale bars represent 200 µm, two samples from each tissue for each age group were stained for ALK with one core represented for each age group).

other than in the pituitary (3.53, *n* = 283) and testis (3.08, *n* = 361) (Supplemental Table 2). Median *ALK* RNA TPM for neoplastic histology outliers and for normal histologies in reference to median *ALK* RNA TPM in neuroblastoma were plotted as box plots of median log 2 transformed TPM + 1 to account for tissues with a TPM of 0 (Fig. 1A, B).

Next, we sought to characterize ALK protein expression in 55 human neuroblastoma tumors and in 43 normal pediatric tissues using immunohistochemistry (IHC) staining on tissue microarrays (TMAs) with the clinically validated and FDA approved D5F3 anti-ALK antibody, known to bind to the cytoplasmic kinase domain of ALK[31,36]. To evaluate expression and heterogeneity, we employed a H-score calculated by multiplying staining intensity (0 = negative, 1 = equivocal/ uninterpretable, 2 = weakly positive, 3 = strongly positive) by the percent of stained cells resulting in a scale of 0-300. The average ALK H-score for each tumor was used for analysis when duplicate or triplicate cores were provided for a single tumor. ALK protein expression was present in most human neuroblastoma tumor samples with an average H-score of 144.71 (*n* = 55 tumors) without any significant differences in H-score when tumor samples were grouped by *MYCN* amplification status (amplified and non-amplified), *ALK* status (wild-type [WT] and mutated), sex, age (<12 months old, 12 to 18 months old, 18 month to 5 years old, >5 years old), differentiation (poorly

differentiated and differentiated), histology (unfavorable and favorable), ploidy (diploid and hyperdiploid), mitosis-karyorrhexis index (MKI; high, intermediate, and low), time of biopsy (diagnostic, post-chemotherapy, and at relapse), revised International Neuroblastoma Risk Group Staging System (INRGSS) stratification[37], International Neuroblastoma Risk Group (INRG) stage, or from patients with a diagnosis of ganglioneuroblastoma (average H-score 133.14, *n* = 5 tumors, Fig. 2A, B, Supplemental Table 3). The average H-score was 1.69 in human placenta (*n* = 20 cores), 0.23 in human adrenal cortex (*n* = 2 cores), 0 in human adrenal medulla (*n* = 2 cores), and 0 in human tonsil (*n* = 2 cores, Fig. 2A, B, Supplemental Table 3). A 20x magnified image of a diagnostic neuroblastoma core with an H-score of 142.33 was provided for reference of near-average ALK staining in the patient tumor TMA (Fig. 2C). ALK protein expression was absent in nearly all normal TMA samples other than equivocal ALK staining in four pituitary samples (1+ staining in a subset of cells), 2 adrenal medulla samples (1+ staining in a subset of cells), and a single smooth muscle (1 +), single ureter smooth muscle (1 +), and single spinal cord sample (rare 1+ cells) (Fig. 2D, Supplemental Table 4). To correlate ALK IHC staining in the highest *ALK* RNA-expressing normal tissues, individual TMA slides for pituitary and testis were provided. ALK staining was equivocal in pituitary TMA samples from age groups 0–2 years and 2–8

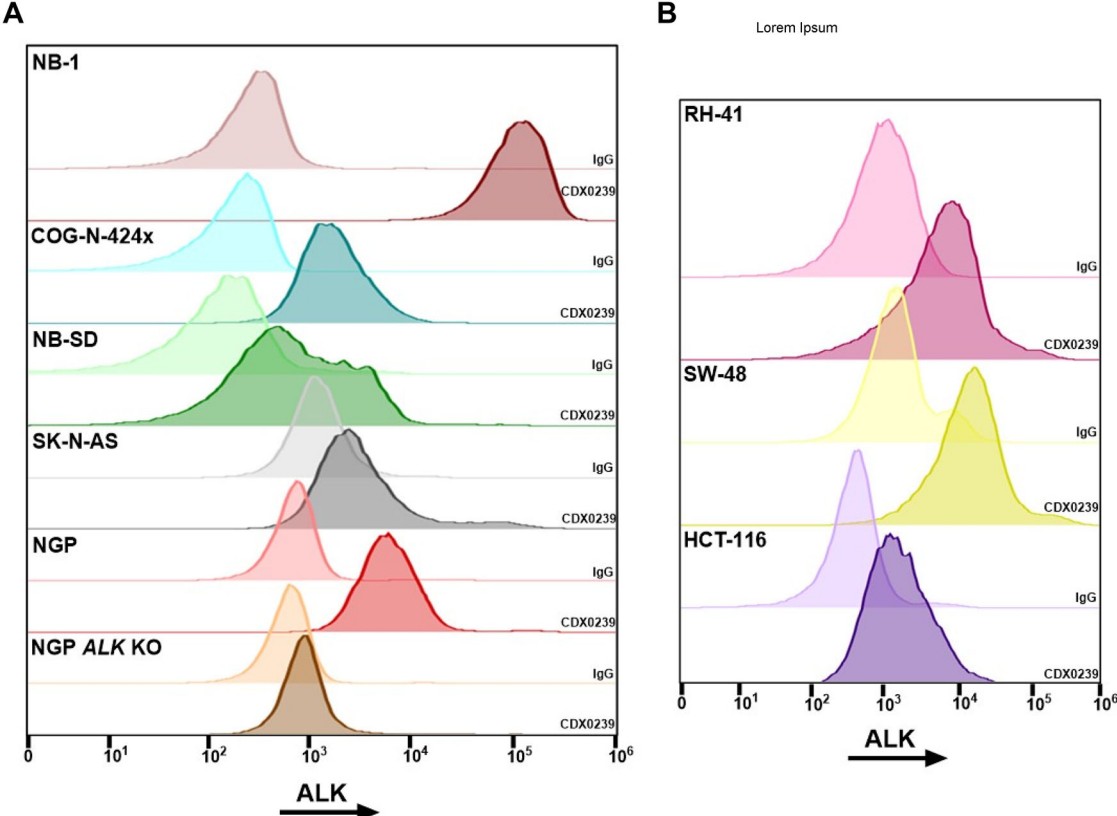

**Fig. 3 | ALK is expressed on the cell surface in neuroblastoma, fusion-positive rhabdomyosarcoma, and colorectal carcinoma xenograft models. A** Flow cytometry histograms for ALK using CDX0239 compared to IgG control in neuroblastoma xenograft models NB-1, COG-N-424x, NB-SD, and SK-N-AS, and in neuroblastoma cell lines NGP *ALK* wild-type (WT) and NGP *ALK* knockout (KO). **B** Flow cytometry histograms for ALK using CDX0239 compared to IgG control in fusion-positive rhabdomyosarcoma xenograft model RH-41 and in colorectal carcinoma xenograft models SW-48 and HCT-116. Represented data has been validated with at least 2 independent experiments from different biological xenograft samples derived from individual mice injected with the corresponding cell lines. Geometric means are located within the Source Data file. Source data are provided as a Source Data file.

years (1+ staining in a subset of cells), with absent ALK staining in pituitary TMA samples from age groups 8–12 years and 12–18 years, and ALK staining was absent in the testis TMA samples in all age groups (Fig. 2E).

### ALK is expressed on the cell surface of neuroblastoma, fusion-positive rhabdomyosarcoma, and colorectal carcinoma xenograft models

We selected our lead candidate murine antibody from a panel of generated ALK-directed monoclonal antibodies based on the highest binding affinity via bio-layer interferometry[38,39]. We then humanized this antibody via homology remodeling, and confirmed cross-species reactivity with murine surface ALK determined via flow cytometry with CDX0239 on spontaneously occurring murine neuroblastoma tumors (1331, 13387, 13389) derived from the murine Th-MYCN model[40] (Supplemental Fig. 1). We confirmed ALK cell surface expression via flow cytometry using CDX0239 across four neuroblastoma xenograft models representative of high-risk patients seen in the clinic, an ALK-expressing neuroblastoma cell line with corresponding genetic *ALK* knockout (KO), a fusion-positive rhabdomyosarcoma xenograft model, and two colorectal carcinoma xenograft models. ALK surface expression was compared to IgG control and quantified with fluorescence intensity geometric means. The non-specific binding of the IgG control varied between xenograft models but was consistent within each xenograft model, enabling the use of the ALK:IgG geometric mean as an effective quantitative representation of ALK surface expression. Among neuroblastoma xenograft models, ALK surface

expression was highest in NB-1 (*ALK*-amplified, *MYCN*-amplified; geometric mean 668.7x IgG control), followed by COG-N-424x (*ALK* WT, *MYCN*-amplified; geometric mean 27.7x IgG control), NB-SD (*ALK* F1174L mutated, *MYCN*-amplified, *TP53* mutated; geometric mean 15.0x IgG control), and SK-N-AS (*ALK* WT, *MYCN* non-amplified; geometric mean 3.5x IgG control). The neuroblastoma cell line NGP demonstrated moderate ALK surface expression (*ALK* WT, *MYCN* non-amplified; geometric mean 8.6x IgG control) and there was minimal detection of expression in the NGP *ALK* KO cell line (geometric mean 1.6x IgG control; Fig. 3A, Supplemental Table 5). ALK surface expression compared to IgG control was moderate in fusion-positive rhabdomyosarcoma xenograft model RH-41 (geometric mean 7.02x IgG control), and moderate in colorectal carcinoma xenograft models SW-48 (geometric mean 9.76x IgG control) and HCT-116 (geometric mean 5.59x IgG control, Fig. 3B, Supplemental Table 5). An example of the utilized gating strategy after depletion of mouse myeloid, endothelial, and stromal cells is included for reference (Supplemental Fig. 2).

### CDX0239 demonstrates ALK surface expression level-dependent internalization kinetics with localization to the lysosome

To characterize ALK surface binding and internalization kinetics in neuroblastoma cell lines with varying levels of ALK expression, we conjugated CDX0239 to Fabfluor-pH Orange labeling dye designed for labeling Fc-containing antibodies with a pH sensitive fluorophore. CDX0239 demonstrated internalization in an ALK surface expression level-dependent manner. NB-1 cells with high ALK surface expression

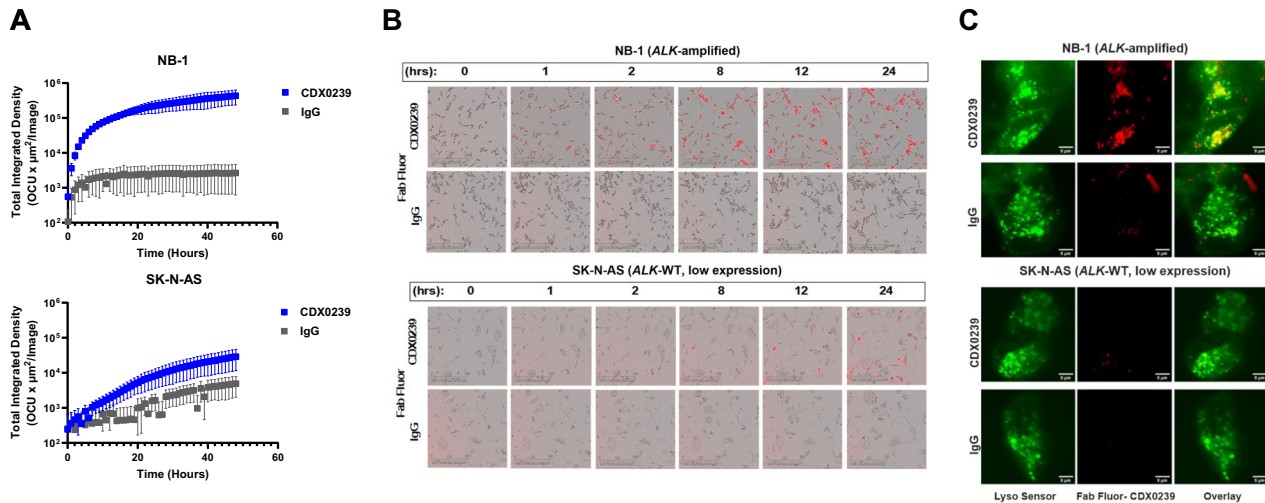

**Fig. 4 | CDX0239 is internalized and localizes to the lysosome in ALK-expressing neuroblastoma cell lines. A** Quantification of internalized Fabfluor-conjugated CDX0239 (blue line with squares) compared to Fabfluor-conjugated IgG control (gray line with squares) in *ALK*-amplified NB-1 and in *ALK* WT low surface-expression SK-N-AS plotted as total integrated density mean ± standard deviation over 48 h with n = 5 technical replicates for the IgG condition in NB-1 and *n* = 6 technical replicates for all other conditions. Sample size and datasets are located within the Source Data File. Source data are provided as a Source Data File. **B** Pictographs of the internalization of Fabfluor-conjugated CDX0239 compared to Fabfluor-conjugated IgG in NB-1 and SK-N-AS over 24 h. **C** Lattice light-sheet microscopy images of NB-1 and SK-N-AS cells treated with LysoSensor 3 h after treatment with Fabfluor-conjugated CDX0239 or Fabfluor-conjugated IgG with overlay. Represented data has been validated with at least 2 independent experiments.

due to underlying gene amplification exhibited logarithmic internalization kinetics of CDX0239 and showed internalization within 2 h and near maximal internalization by 24 h (Fig. 4A, B). SK-N-AS cells with low ALK surface expression exhibited minimal internalization with linear internalization kinetics of CDX0239, with a subset of cells demonstrating internalization within 24 h (Fig. 4A, B). NB-SD and IMR-32 cells with moderate ALK surface expression demonstrated linear internalization kinetics of CDX0239 within 8 h (Supplemental Fig. 3A, B). We further confirmed the lysosomal localization of CDX0239 conjugated to Fabfluor-pH Red utilizing a lysosome-specific LysoSenor Green. Fluorescent images obtained using a custom-built lattice light-sheet microscope showed colocalization of CDX0239 with LysoSensor green after 3 h, most prominently in NB-1 as indicated by yellow puncta on overlay image (Fig. 4C).

### CDX0239-PBD demonstrates potent and target-mediated cytotoxicity in neuroblastoma cell lines

PBD was selected as the ADC payload based on known sensitivities of 11 neuroblastoma cells lines that included a screen of tubulin inhibitors, DNA binding agents, and topoisomerase I inhibitors with DNA binding agents demonstrating the highest potency and low rates of resistance[41]. CDX0239-PBD was synthesized via a three-branched linker conjugation to glutamine 295 (Q295) followed by click reaction for conjugation of the PBD-dimer payload with a valine-alanine (VA) cleavage site, a PEG4 linker, and a dibenzocyclooctyne group (DBCO-PEG4-VA-PBD) (Fig. 5A). CDX0239-PBD demonstrated ALK surface expression level-dependent cytotoxicity in neuroblastoma cell lines 5 days after treatment. $IC_{50}$ values after treatment with CDX0239-PBD were 16.1 pM vs 342 pM with free PBD in high surface ALK-expressing NB-1, 22.5 pM vs 619 pM with free PBD in moderate surface ALK-expressing NB-SD, and 5,070 pM vs 64.3 pM with free PBD in low surface ALK-expressing SK-N-AS (Supplemental Table 6). Viability of CDX0239-PBD-treated cell lines was significantly lower with treatments of CDX0239-PBD compared to free PBD at concentrations as low as 1 pM in NB-1, 10 pM in NB-SD, and this effect was not observed in SK-N-AS. CDX0239 without PBD payload did not lead to a decrease in cell viability in comparison to untreated controls in any neuroblastoma

cell lines (Fig. 5B). NGP cell line $IC_{50}$ values 5 days after treatment with CDX0239-PBD was 42.79 pM vs 31.08 pM with free PBD in NGP *ALK* WT and 24.40 μM (24,400,000 pM) vs 32.73 pM with free PBD in NGP *ALK* KO (Fig. 5C, Supplemental Table 6).

To support the mechanism of CDX0239-PBD-induced cytotoxicity to be free intracellular PBD-induced DNA damage and subsequent apoptosis, we measured caspase 3 and 7 activity. Treatment of neuroblastoma cell lines with CDX0239-PBD led to significantly higher levels of caspase activity as compared to treatment with free PBD at concentrations as low as 1 pM in high surface ALK-expressing NB-1, 10 pM in moderate surface ALK-expressing NB-SD, and this effect was not observed in low surface ALK-expressing SK-N-AS (Fig. 5D). Lastly, to demonstrate that CDX0239-PBD requires binding of surface ALK for internalization and downstream intracellular delivery of free PBD, and to demonstrate the stability of the linker, high surface ALK-expressing NB-1 cells were cultured with 8 pM of CDX0239-PBD alone and in combination with increasing concentrations of CDX0239 without PBD payload for 5 days as a direct competitor for surface ALK binding. There was a significant increase in NB-1 cell viability after 5 days of treatment when concentrations of CDX0239 without PBD payload as low as 7.5 nM (7500 pM) were added to co-culture with 8 pM of CDX0239-PBD (Fig. 5E).

### CDX0239-PBD Demonstrates Potent Efficacy with Complete and Sustained Responses in ALK-Expressing Neuroblastoma Xenograft Models

We next determined the in vivo anti-tumor activity of CDX0239-PBD in conventional and patient-derived xenograft models representing different patient subpopulations of neuroblastoma, including varying ALK cell surface expression and *ALK* mutation/amplification status. Neuroblastoma xenograft models included NB-1 (*ALK*-amplified, *MYCN* amplified), NB-SD (moderate surface ALK-expression, *ALK* F1174L mutated, *MYCN* amplified, *TP53* mutated), COG-N-424x (moderate surface ALK-expression, *ALK* WT, *MYCN* amplified), SK-N-AS (low surface ALK expression, *ALK* WT, *MYCN* non-amplified, *TP53* mutated), NGP *ALK* WT (moderate surface ALK-expression, *MYCN* amplified, *TP53* mutated) and NGP *ALK* KO (ALK null, *MYCN* amplified, *TP53* mutated)

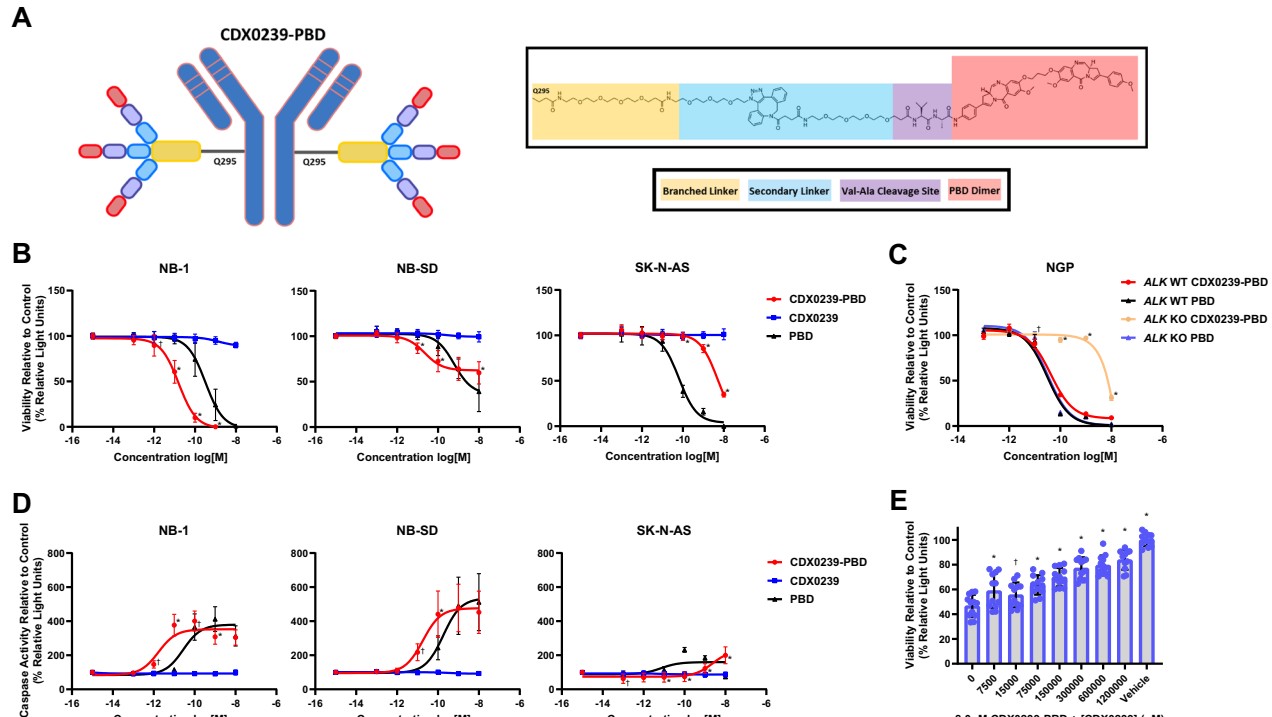

**Fig. 5 | CDX0239-PBD demonstrates potent in vitro cytotoxicity in an ALK surface expression level-dependent manner in neuroblastoma cell lines.**
**A** Structural representation of CDX0239-PBD with humanized anti-ALK antibody (CDX0239), linkers, cleavage site, and PBD payload. **B** Viability of NB-1, NB-SD, and SK-N-AS cell lines 5 days after treatment with CDX0239-PBD (red line with circles), free PBD (black line with triangles), and CDX0239 (blue line with squares) as compared to untreated controls quantified as percent relative light units ($\dagger P < 0.05$, $*P < 0.0001$ vs free PBD treatment using a using a three-parameter log vs response analysis with $n = 12$ technical replicates for the CDX0239-PBD condition in all cell lines and the CDX0239 condition in NB-1 and NB-SD, and $n = 9$ technical replicates for the CDX0239 condition in SK-N-AS and for the free PBD condition in all cell lines).
**C** Viability of NGP *ALK* wild-type (WT) 5 days after treatment with CDX0239-PBD (red line with circles) and free PBD (black line with triangles) and of NGP *ALK* knockout (KO) 5 days after treatment with CDX0239-PBD (peach line with circles) and free PBD (lavender line with triangles) as compared to untreated controls quantified as percent relative light units ($\dagger P < 0.05$, $*P < 0.0001$ vs *ALK* WT

CDX2039-PBD using a using a three-parameter log vs response analysis with $n = 3$ technical replicates per condition). **D** Caspase activity in NB-1, NB-SD, and SK-N-AS cell lines after treatment with CDX0239-PBD (red line with circles), free PBD (black line with triangles), and CDX0239 (blue line with squares) as compared to untreated controls quantified as percent relative light units ($\dagger P < 0.05$, $*P < 0.0001$ vs free PBD treatment using a using a three-parameter log vs response analysis with $n = 12$ technical replicates for all conditions in NB-1 and NB-SD and $n = 9$ technical replicates for all conditions in SK-N-AS). **E** NB-1 cell line viability 5 days after treatment with 8 pM of CDX0239-PBD and increasing concentrations of co-treatment with CDX0239 as compared to controls treated with cell culture media only (vehicle) quantified as percent relative light units ($\dagger P < 0.05$, $*P < 0.0001$ vs 0 pM CDX0239 using a two-away ANOVA analysis with Dunnett's multiple comparison test with $n = 12$ technical replicates per condition). All data are plotted as mean ± standard deviation. Represented data has been validated with at least 3 independent experiments. Sample size, datasets, and exact *p*-values are located within the Source Data file. Source data are provided as a Source Data file.

(Supplemental Table 7). A treatment schedule of 1 mg/kg CDX0239-PBD was used initially at enrollment and repeated at week 1 and at week 2 for a total of 3 doses. Treatment of neuroblastoma xenograft models with CDX0239-PBD led to maintained complete response in NB-1, NB-SD, COG-N-424x, and NGP *ALK* WT, and a partial response with subsequent tumor growth progression in SK-N-AS (Fig. 6A). Non-target IgG and CDX0239 controls were utilized to demonstrate lack of non-target human IgG and humanized anti-ALK anti-tumor activity, respectively. CDX0239 deglycosylation via asparagine to alanine mutation or enzymatic digestion controls were utilized to maximize antibody-dependent cellular cytotoxicity (ADCC)[42]. CDX0239 deglycosylation via asparagine to alanine mutation with conjugated L6 linker control was utilized to demonstrate lack of linker-induced anti-tumor activity. All neuroblastoma xenograft tumors treated with IgG, normal saline, CDX0239, or modified CDX0239 controls led to tumor volume progression (Supplemental Fig. 4A). There was 100% overall survival in neuroblastoma xenograft models NB-1, NB-SD, COG-N-424x, and NGP *ALK* WT, and 40% overall survival in SK-N-AS after treatment with CDX0239-PBD (Fig. 7A). There was 0% overall survival in neuroblastoma xenograft models treated with each control condition (Supplemental Fig. 5A). Treatment of NB-1 xenograft models with

a single 1 mg/kg dose of CDX0239-PBD led to a maintained complete response (Fig. 6C) and 100% overall survival (Fig. 7C). Treatment of NGP *ALK* KO xenografts with CDX0239-PBD led to rapid tumor growth in 3 xenografts and an initial transient response with subsequent rapid tumor growth in 2 xenografts. There were no significant differences in average tumor volume when treated with CDX0239-PBD compared to normal saline vehicle (Fig. 6D) with an overall survival rate of 0% (Fig. 7D). No significant weight loss or other signs of toxicity were observed (Supplemental Fig. 6A, C).

## CDX0239-PBD demonstrates potent efficacy with complete and sustained responses in ALK-expressing colorectal carcinoma and fusion-positive rhabdomyosarcoma xenograft models

Next, to broaden the therapeutic application of an ALK-directed ADC in additional ALK-expressing malignancies, we treated xenografts of HCT-116 (colorectal carcinoma, derived from 44-year-old male patient, *KRAS* c.38 G > A, G13D[43,44] and *PIK3CA* c.3140 A > G, H1074R[43] mutated) and SW-48 (colorectal carcinoma, derived from an 83-year-old female patient, *CTNNB1* c.98 C > A, S33Y[43] and *APC* c.8140 C > T, R2714C[43] mutated), and a xenograft of RH-41 (PAX3::FOXO1 fusion-positive rhabdomyosarcoma) with CDX0239-PBD using an identical dosing

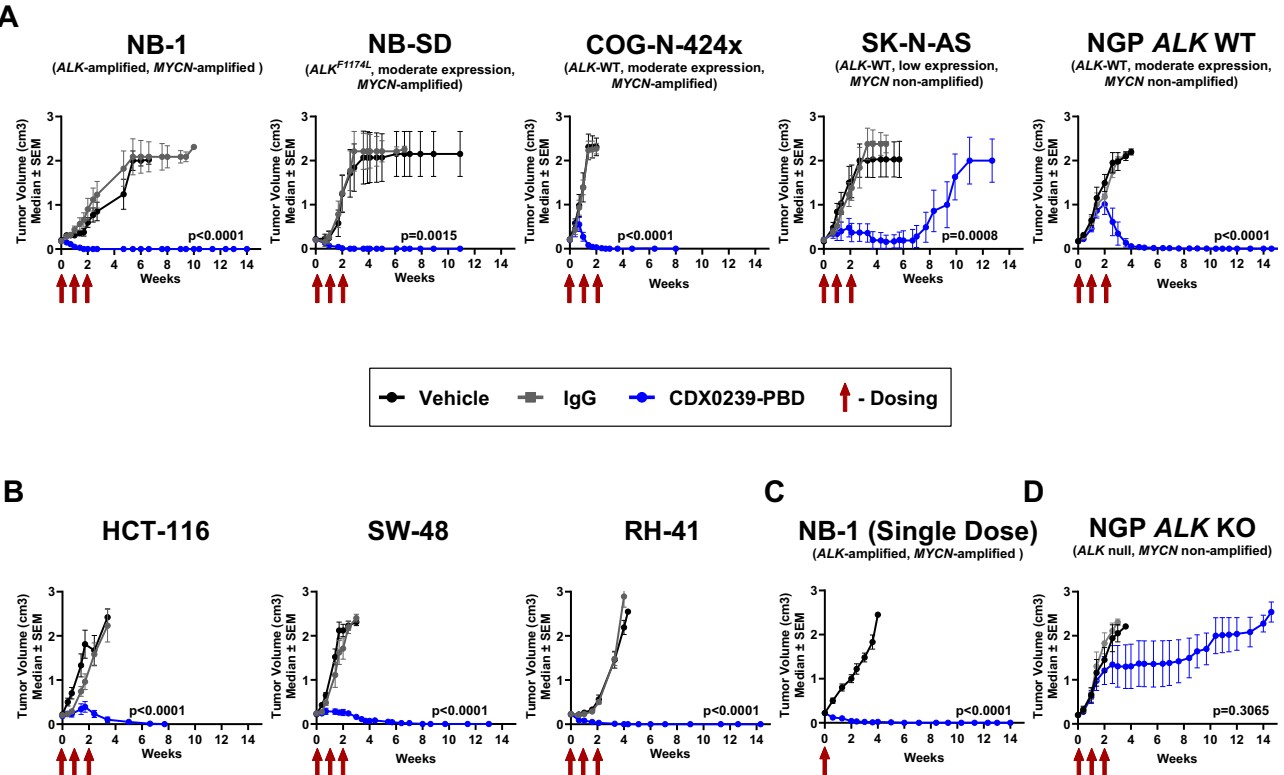

**Fig. 6 | CDX0239-PBD demonstrates potent in vivo antitumor activity in ALK-expressing xenograft models. A** Average tumor volume in neuroblastoma xenograft models NB-1, NB-SD, COG-N-424x, SK-N-AS, and NGP *ALK* wild-type (WT) with 3 weekly treatments of 1 mg/kg CDX0239-PBD (blue line with circles) compared to IgG (gray line with squares) and normal saline vehicle (black line with circles). **B** Average tumor volume of colorectal carcinoma xenograft models HCT-116 and SW-48 and fusion-positive rhabdomyosarcoma xenograft model RH-41 with 3 weekly treatments of 1 mg/kg CDX0239-PBD (blue line with circles) compared to IgG (gray line with squares) and normal saline vehicle (black line with circles). **C** Average tumor volume in neuroblastoma xenograft model NB-1 with a single 1 mg/kg dose of CDX0239-PBD (blue line with circles) compared normal saline vehicle (black line with circles). **D** Average tumor volume in neuroblastoma xenograft model NGP *ALK* knockout (KO) with 3 weekly treatments of 1 mg/kg CDX0239-PBD (blue line with circles) compared to IgG (gray line with squares) and normal saline vehicle (black line with circles). All data are plotted as mean ± standard deviation with n = 5 mice per treatment condition for all experiments, and each experiment was completed once. All *p*-values represent CDX0239-PBD vs vehicle tumor volumes using a square root transformation analysis with linear mixed effects model. Exact *p*-values and datasets are located within the Source Data file. Source data are provided as a Source Data file.

schedule of 1 mg/kg CDX0239-PBD initially at enrollment and repeated at week 1 and at week 2 for a total of 3 doses (Supplemental Table 7). Each colorectal carcinoma and fusion-positive rhabdomyosarcoma xenograft model showed a maintained complete response after treatment with CDX0239-PBD (Fig. 6B), and all models treated with IgG, normal saline, CDX0239, or modified CDX0239 controls led to tumor volume progression (Supplemental Fig. 4B). There was 100% overall survival in each colorectal carcinoma and fusion-positive rhabdomyosarcoma model treated with CDX0239-PBD (Fig. 7B) and 0% overall survival in all models treated with controls (Supplemental Fig. 5B). No significant weight loss or other signs of toxicity were observed (Supplemental Fig. 6B).

### CDX0239-PBD is active in a neuroblastoma tumor model with de novo resistance to lorlatinib

To establish CDX0239-PBD as a therapeutic modality for patients with activating *ALK* mutations who either have de novo or acquired resistance to small enzymatic molecular ALK inhibitors, we tested CDX0239-PBD in the NB-SD xenograft model (moderate surface ALK-expression, *ALK* F1174L-mutated, *MYCN* amplified, *TP53* mutated). We treated with either 1 mg/kg CDX0239-PBD at enrollment and repeated at week 1 and at week 2 for a total of 3 doses or daily 10 mg/kg lorlatinib[45] for 3 weeks with the CDX0239-PBD treatment group. Notably, NB-SD tumors treated with daily lorlatinib demonstrated progressive tumor growth and 0% overall survival, while NB-SD tumors treated with CDX0239-PBD showed maintained complete responses

with 100% survival over a total observation period of 9 weeks (Fig. 8A–D).

### CDX0239-PBD leads to DNA damage and induction of apoptosis in ALK-expressing xenografts

To confirm that the efficacy of CDX0239-PBD in vivo was due to DNA crosslinking with subsequent DNA damage and induction of apoptosis due to free intracellular PBD, we analyzed NB-SD tumor samples treated with a single 10 mL/kg normal saline treatment (vehicle) or with a single 1 mg/kg dose of CDX0239-PBD and collected tumor samples at 3 and 7 days after treatment. CDX0239-PBD induced increased expression of the DNA damage marker phosphorylated H2Ax (γH2Ax), and increased expression of markers of apoptosis cleaved caspase-3 (cCaspase-3) and cleaved poly(ADP-ribose) polymerase (cPARP) compared to vehicle control 7 days after treatment. CDX0239-PBD did not affect levels of phosphorylated ALK (pALK1604), and we observed a qualitative decrease in total ALK in NB-SD tumors treated with a single dose of CDX0239-PBD after 7 days. NB-SD tumors did not express the cleaved N-terminus of ALK that has been observed in other ALK-expressing neuroblastoma cell lines and xenografts[46] (Fig. 8E).

### Discussion

Immunotherapy for cancer has experienced a multitude of breakthroughs in the last 20 years. ADC trials have tripled in the past 5 years[32,47], but only a few ADCs have been approved for use in the

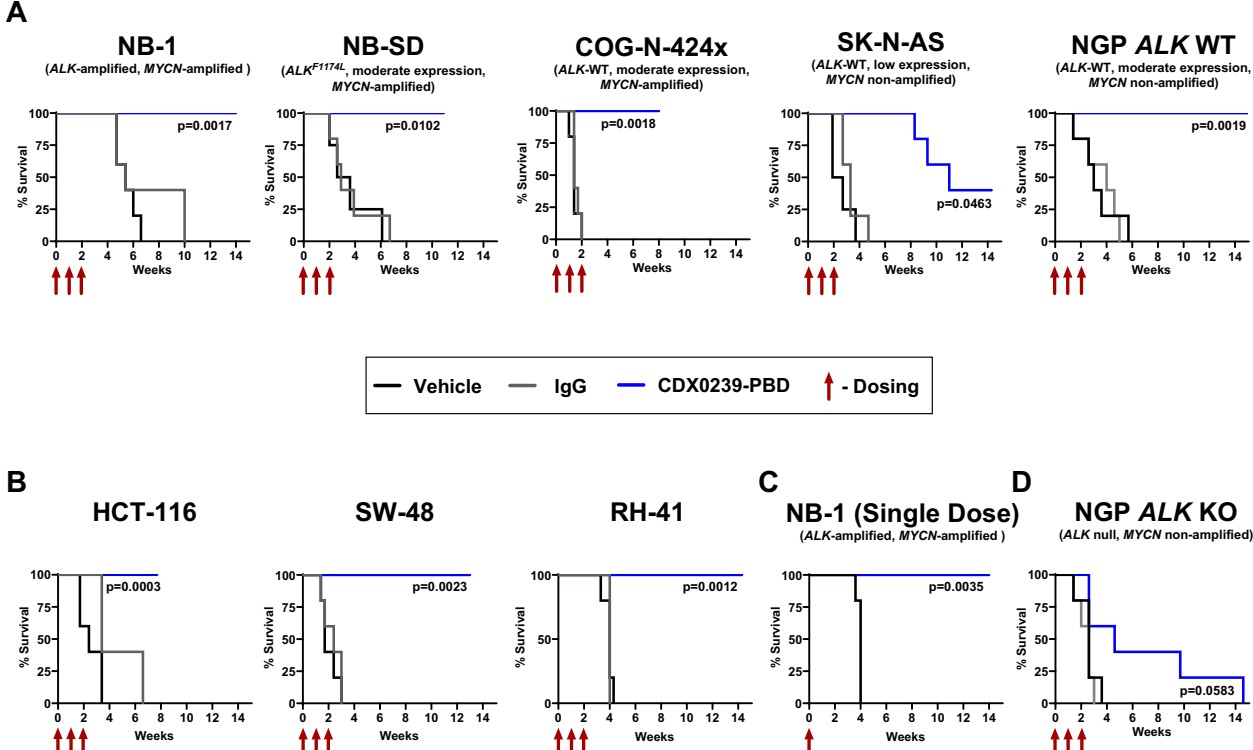

**Fig. 7 | Treatment with CDX0239-PBD leads to survival in ALK-expressing xenograft models. A** Overall survival of neuroblastoma xenograft models NB-1, NB-SD, COG-N-424x, SK-N-AS, and NGP *ALK* wild-type (WT) with 3 weekly treatments of 1 mg/kg CDX0239-PBD (blue line) compared to IgG (gray line) and normal saline vehicle (black line). **B** Overall survival of colorectal carcinoma xenograft models HCT-116 and SW-48 and fusion-positive rhabdomyosarcoma xenograft model RH-41 with 3 weekly treatments of 1 mg/kg CDX0239-PBD (blue line) compared to IgG (gray line) and normal saline vehicle (black line). **C** Overall survival of

neuroblastoma xenograft model NB-1 with a single 1 mg/kg dose of CDX0239-PBD (blue line) compared to normal saline vehicle (black line). **D** Overall survival of neuroblastoma xenograft model NGP *ALK* knockout (KO) with 3 weekly treatments of 1 mg/kg CDX0239-PBD (blue line) compared to IgG (gray line) and normal saline vehicle (black line). Each treatment condition includes a sample size of *n* = 5 mice and each experiment was completed once. All *p*-values represent CDX0923-PBD vs vehicle tumor volumes using Kaplan-Meier analysis. Exact *p*-values and datasets are located within the Source Data file. Source data are provided as a Source Data file.

United States and Europe, primarily for lymphoma and leukemia[48,49]. Similar progress has not been made in the treatment of solid tumors with an ADC approach partly due to the difficulty of identifying acceptable targets with expression restricted to tumor and not in normal tissues. Our analysis of publicly available RNA expression data for over 12,000 tumor samples supports nominating ALK as an exemplary lineage-restricted ADC target across multiple childhood and adult malignancies. We further demonstrate that native ALK protein is widely and differentially expressed across an expansive panel of neuroblastoma patient tumors representative of the clinical and molecular features observed in clinical practice, further validating prior observations that ALK is among the most abundant proteins in neuroblastoma cell lines and xenograft models[50]. Furthermore, previously published studies have demonstrated overexpression of native *ALK* RNA and protein in aggressive subsets of metastatic colorectal carcinoma[6], malignant melanoma[7,11], fusion-positive rhabdomyosarcoma[4,5], and breast and ovarian carcinomas in need of novel therapies[9,11]. Importantly, we show minimal *ALK* RNA and protein expression in normal tissues other than the immunologically privileged pituitary and testis where foreign antigens such as ALK can be tolerated due to the blood-brain barrier (BBB) and blood-testis barrier (BTB)[51], respectively.

These data support the development of an ALK-directed ADC approach. We selected highly potent PBD dimers[35] as our ADC payload for maximal efficacy, minimal off-target toxicities, and a potentially large therapeutic window while remaining cognizant of its clinical liabilities[47]. Loncastuximab tesirine is a PBD-based ADC targeting CD19

refractory diffuse large B cell lymphoma (DLBCL) that has received its first FDA approval, demonstrating the potential for PBD-conjugated ADCs[52]. CDX0239-PBD utilizes a humanized ALK-directed antibody with high affinity for the extracellular domain which allows more readily for clinical implementation with decreased risks of immunogenicity and adverse events compared with our previous chimeric ALK-directed ADC[29,34]. Additionally, ADCs enable bystander cytotoxicity to tumor cells with low surface target antigen expression that leads to activity in the solid tumor microenvironment (TME) and extracellular matrix (ECM), compartments that naturally prevent localization of large molecules[53]. This bystander effect permits ADCs to have unique immune-enhancing properties that overcome the inherent structural and spatial barriers to drug delivery in the TME, maximizing the translational potential of an ALK-directed ADC for aggressive and often-lethal solid tumors[54]. In particular, these features include the release of damage-associated molecular patterns (DAMPs) after inducing tumor cell death that leads to dendritic cell activation to facilitate antigen uptake and migration to lymph nodes, ADC-mediated reduction of regulatory T cells (Tregs), augmentation of major histocompatibility complex (MHC) class I expression to promote activity of cytotoxic T cells, and enhancement of leukocyte infiltration, expansion, and interferon gamma (IFNγ) production[55]. Additionally, a therapeutic challenge with large molecules is localization across the BBB to the CNS for the treatment of brain metastases, a rare but dismal challenge for neuroblastoma[56], and primary CNS malignancies. Importantly, there is selective disruption of the BBB in the setting of some tumors with CNS metastases[57]. Consistent with these findings, a

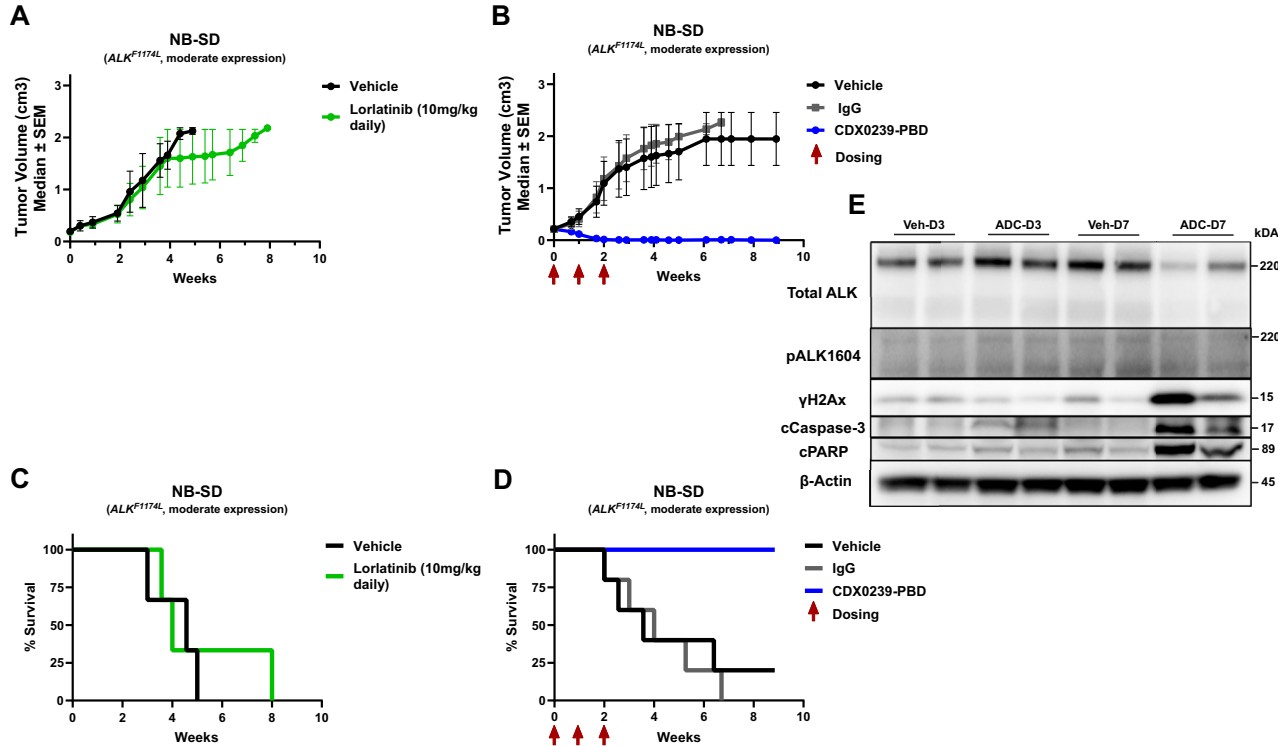

**Fig. 8 | CDX0239-PBD demonstrates potent in vivo efficacy in ALK-expressing and lorlatinib-resistant neuroblastoma NB-SD xenograft model via intracellular delivery of PBD with subsequent DNA damage and apoptosis. A** Average tumor volume with daily lorlatinib treatments at 10 mg/kg (green line with circles) compared to normal saline vehicle (black line with circles). **B** Average tumor volume with 3 weekly treatments of 1 mg/kg CDX0239-PBD (blue line with circles) compared to IgG (gray line with squares) and normal saline vehicle (black line with circles). **C** Survival of xenograft models with daily lorlatinib at 10 mg/kg (green line) compared to normal saline vehicle (black line). **D** Survival of xenograft models with 3 weekly treatments of 1 mg/kg CDX0239-PBD (blue line) compared to IgG (gray line) and normal saline vehicle (black line). **E** Western Blot analysis of total ALK, phosphorylated ALK (pALK1604), phosphorylated H2Ax (γH2Ax), cleaved caspase 3 (cCaspase-3), and cleaved PARP (cPARP) in NB-SD xenograft models 3 and 7 days after treatment with 10 mL/kg normal saline vehicle (Veh-D3 and Veh-D7, respectively) or 1 mg/kg CDX0239-PBD (ADC-D3 and ADC-D7, respectively). Average tumor volumes are plotted as mean ± standard deviation with $n = 5$ mice per treatment condition for all experiments, and each experiment was completed once. Represented data for Western blot experiments has been validated with 2 independent experiments using the same biologic replicate tumor samples. Sample size, datasets, and unprocessed western blot scans are located within the Source Data file. Source data are provided as a Source Data file.

HER2-directed ADC demonstrated complete and partial intracranial responses in HER2 low-expressing breast cancer with brain metastases[58], suggesting the potential for ADC anti-tumor activity in the CNS.

Target antigen expression on cancer cells significantly impacts the effectiveness of ADCs. The antigen expression threshold above which ADCs are expected to be active is not fully understood and will likely vary by ADC due to antigen binding and payload. While high expression is desirable for the success of chimeric antigen receptor (CAR) T cell therapies[59,60], our data demonstrates robust and sustained anti-tumor activity in models with modest ALK expression levels. We show complete maintained responses to CDX0239-PBD in NB-SD, COG-N-424x, and NGP, xenografts that do not harbor *ALK* gene amplification and exhibit ALK surface expression representative of the majority of neuroblastoma tumors and relevant xenograft models[60]. This is supported by the patient TMA IHC data demonstrating significant ALK protein expression across a diverse group of neuroblastoma tumors, suggesting that the vast majority of patients are above the therapeutic threshold of ALK surface expression and could benefit from an ALK-directed ADC. SK-N-AS, which exhibits very low ALK surface expression, did not demonstrate complete and sustained tumor regressions but the transient anti-tumor activity suggests the presence of bystander cytotoxicity and potential to provide adjuvant therapeutic benefit in low ALK-expressing tumors. The NGP *ALK* KO xenografts did not show significant differences in tumor growth when treated with CDX0239-PBD compared to vehicle control, which demonstrates ALK cell surface-expression-dependent efficacy. We postulate that the initial transient response of two out of the five NGP *ALK* KO xenografts was due to non-specific internalization of CDX0239-PBD via binding of the Fc region on the full-length antibody to FCγR,s which has been previously demonstrated in full-length antibody-based ADCs[61,62]. This effect can be addressed with FCγRs blocking antibodies in combination with CDX0239-PBD treatment or by leveraging antibody engineering approaches such as L234A/L235A substitutions to reduce Fc-mediated effector functions[62]. Future studies will investigate additional linker-payload chemistries, including conjugation methods shielding the linker-payload molecule within the Fc region to reduce hydrophobicity, aggregation, and non-specific internalization of ADCs[63].

Target antigen selectivity and linker stability are vital for the efficacy and safety of ADCs. We show that CDX0239-PBD induces antibody-receptor internalization and localization to the lysosome where the pH-sensitive L6-linker is cleaved leading to enhanced intracellular delivery of PBD in an ALK surface expression-dependent manner. Importantly, the lack of activity of CDX0239-PBD in an *ALK* KO cell line and when in competition for surface ALK with unconjugated CDX0239 shows that CDX0239-PBD requires antigen binding and internalization for potent activity and demonstrates stability of the linker. This is further supported in the NPG *ALK* KO xenograft models where we observe no significant differences in tumor growth compared to vehicle control and with our numerous in vivo controls. There is no anti-tumor effect from unconjugated CDX0239 alone and we did

not observe evidence of phosphorylated ALK attenuation after treatment with CDX0239-PBD in NB-SD xenografts. This model harbors an activating *ALK* mutation, which confers constitutive ALK activation in the absence of ligand binding[64]. Regarding tumors without an activating *ALK* mutation, we postulate that unconjugated CDX0239 does not have a significant inhibitory effect on ALK signaling as we would expect to observe anti-tumor activity in the *ALK*-amplified NB-1 cell line and xenograft model as previously demonstrated with small molecular inhibition of ALK[25]. CDX0239 deglycosylation via asparagine to alanine mutation or enzymatic digestion to maximize ADCC by enhancing antibody binding to FcγRIIIA (CD16) on leukocytes[42] did not show activity, suggesting no evidence of ADCC with CDX0239 or CDX0239-PBD. CDX0239 deglycosylation via asparagine to alanine mutation with conjugated L6 linker did not show activity, suggesting no evidence of linker-induced cytotoxicity. Lastly, there were no significant adverse events or toxicity after treatment with CDX0239-PBD, a molecule with antibody cross-reactivity between mouse and human ALK, further demonstrating antigen selectivity and linker stability of CDX0239-PBD and facilitating preclinical testing and drug development endeavors.

The observed decrease in total ALK correlating with increased markers of DNA damage (γH2AX[65]) and apoptosis (cleaved caspase-3[66] and cleaved PARP[67]) in the NB-SD xenograft models 7 days after treatment with CDX0239-PBD suggests that ALK is being exploited as an intracellular delivery mechanism for CDX0239-PBD via antibody-receptor internalization. Importantly, this mechanism was confirmed at clinically feasible dosing of 1 mg/kg[33,68] in the *ALK*-mutated NB-SD xenograft model which has de novo resistance to lorlatinib, likely due to an underlying pathogenic *TP53* mutation. We demonstrate potent and sustained efficacy of a single dose of CDX0239-PBD in the NB-1 xenograft model, which harbors high-level amplification of both *ALK* and *MYCN*, a combination with portends for particularly dismal patient outcomes[20,69,70]. Similar anti-tumor activity is observed in NGP and COG-N-424x xenografts, *ALK* WT models with varying levels of *ALK* overexpression and *MYCN* status, as observed in patient tumors. These results highlight that an ALK-directed ADC will have activity across a broad range of neuroblastoma patients, including those with an activating *ALK* mutation who do not benefit from lorlatinib.

We observed potent and sustained anti-tumor activity with CDX0239-PBD in surface ALK-expressing colorectal carcinoma xenograft models. HCT-116 harbors oncogenic driver mutations in *KRAS* (c.38 G > A, G13D) and *PIK3CA* (c.3140 A > G, H1074R) and SW-48 harbors mutations in the oncogenic driver *CTNNB1* (c.98 C > A, S33Y) and the tumor suppressor *APC* (c.8140 C > T, R2714C)[43,44]. These results highlight the application of CDX0239-PBD for more aggressive forms of colorectal carcinoma with poor outcomes. Importantly, *ALK* expression has been negatively correlated with relapse-free survival (RFS) in consensus molecular subtype 1 (CMS1) colorectal carcinoma compared to CMS1 tumors without *ALK* expression (hazard ratio of 2.77)[6]. Lastly, we show potent efficacy in an ALK-expressing rhabdomyosarcoma xenograft model RH-41 carrying the PAX3::FOXO1 fusion, a potent and canonical oncogenic driver of this often-lethal childhood sarcoma[71]. These patients have a dismal 5-year event-free survival (EFS) at time of relapse or when presenting with metastatic disease[72], and the cell surface expression of ALK presents a unique treatment vulnerability for patients who continue to receive decades old empiric chemotherapy regimens and lack innovative targeted therapies.

We continue to optimize our ALK-directed ADC platform to enhance activity in solid tumors and mitigate mechanisms of resistance, including antigen loss, instabilities in internalization and recycling, drug efflux and clearance, alterations in signaling pathways, and payload resistance[55]. One such approach is to optimize tumor localization and ALK surface binding by generating ALK-directed ADCs with dimerized heavy chain-only nanobodies, also known as single domain

antibodies[73]. They have the potential for increased potency through enhanced internalization kinetics as these smaller fragments are predicted to have higher diffusion rates and increased TME penetration, a potential advantage for neuroblastoma and other solid malignancies[54,73]. These enhanced properties could decrease the dose required to penetrate tumor cells and reduce potential off-target toxicities. Future studies will evaluate the activity of novel binders and payloads in the Th-MYCN transgenic murine spontaneous tumor model that recapitulates the ECM, stromal cells, pro-inflammatory cells, and immune-mediating cells that influence the diffusion, tumor penetration, and tumor accumulation of ADCs[40,74]. While our data supports PBD as a safe payload for the ALK binder CDX0239 when conjugated to a chemically stable linker, payload diversification with alternate mechanisms of action tailored to tumor sensitivities, such as topoisomerase inhibitors is crucial[41]. Future investigation will encompass conjugation of alternate payloads, modifications to the drug-to-antibody ratios (DARs), and extensive pharmacokinetic studies in non-human primates for Investigational New Drug (IND)-enabling studies.

Clinically relevant antibodies for ALK, as seen for other RTKs such as EGFR[75] and ERBB2[76], have not yet been developed. This work validates ALK as an outstanding immunotherapy target across a spectrum of childhood and adult cancers and provides rationale for the clinical development of a first-in-class ALK-directed ADC. There is real hope that this drug class, often referred to as "precision chemotherapy", could eventually replace some forms of standard chemotherapy.

## Methods

### Ethics statement
Patient samples obtained for TMA creation and annotation with clinical co-variate data was approved by the Children's Hospital of Philadelphia Institutional Review Board Protocol 18-015545, and the study was conducted in accordance with the United States Common Rule. Neuroblastoma tumor cores used to produce the TMAs and samples used to produce PDX models were collected after written, signed and dated informed consent provided by the patient parents/guardians. Tumor cores used to produce the TMAs were biobanked at the Children's Hospital of Philadelphia and no additional compensation was provided. Healthy donors provided informed consent through the University of Pennsylvania Immunology Core. All animal studies were approved by the Children's Hospital of Philadelphia Institutional Animal Care and Use Committee protocol no. IAC 000643 with adherence to the National Institutes of Health guide for Care and Use of Laboratory Animals accredited by the Association for Assessment and Accreditation of Laboratory Animal Care (AAALAC).

### ALK RNA expression in neoplastic and somatic histologies
Neoplastic histology *ALK* RNA expression was determined from data from the Tumor Compendium v11 of UCSC's Treehouse Childhood Cancer Initiative across tumors. Normal histology *ALK* RNA expression was determined from data from the Genotype-Tissue Expression Project v8. The data were generated by polyA selection method and quantified as log2(TPM). Cancer types with outlier RNA expression in *ALK* was determined by calculating percentiles of transcripts per million (TPM). Histologies with median expression within top 95th percentile (median TPM > = 6.56 TPM) or histologies with more than 1 sample in top 99th percentile (more than 1 sample >= 31.04 TPM) of TPM was considered to have an outlier *ALK* expression. Expression was visualized as boxplots which were generated in R version 4.2.3 using ggplot2 function from Tidyverse package version 2.0.0 and were sorted according to median TPM values. Median *ALK* RNA TPM for neoplastic histology outliers and for normal histologies in reference to median *ALK* RNA TPM in neuroblastoma were plotted as box plots of median log 2 transformed TPM + 1 to account for tissues with a TPM of 0.

## Immunohistochemistry staining for ALK

A neuroblastoma TMA was constructed with duplicate punches from formalin-fixed paraffin-embedded human neuroblastoma tumors archived at the Children's Hospital of Philadelphia. IHC was performed with rabbit anti-ALK D5F3 (Cell Signaling, #3633) on formalin-fixed paraffin-embedded TMA slides as previously described[31,77]. D5F3 is an FDA and Clinical Laboratory Improvement Amendments (CLIA)-approved antibody as a standalone test for *ALK* rearrangements in lung cancer and is known to bind to the cytoplasmic kinase domain of ALK[36]. A total of 55 unique human neuroblastoma tumors were analyzed for ALK staining with two cores per tumor in 48 tumors, three cores for one tumor, and a single core analyzed in six tumors. A total of five ganglioneuroblastoma tumors were analyzed with two cores per tumor in four tumors and a single core analyzed for one tumor. Normal pediatric tissue samples included in the human neuroblastoma TMA were scored for comparison to neuroblastoma tumor cores. A total of 20 placenta, 2 tonsil, 2 adrenal cortex, and 2 adrenal medulla specimens were analyzed. Briefly, staining was performed on a Bond Max automated staining system (Leica Biosystems) with Bond Refine polymer staining kit (Leica Biosystems). Anti-ALK D5F3 was used at 1:500 dilution with E20 (Leica Biosystems) retrieval solution for 20 min with slides rinsed and dehydrated through a series of ascending concentrations of ethanol and xylene followed by covering with coverslips. Stained slides were digitally scanned at 20× magnification on an Aperio CS-O slide scanner (Leica Biosystems). TMAs were scored for the most prominent intensity (0–3 with 1 representing equivocal, 2 weak, and 3 strong positive staining) and multiplied by the percent of stained cells resulting in a scale of 0-300. Equivocal staining is defined as minimally present staining of a cell with anti-ALK D5F3 at 1:500 dilution. The average ALK H-score for each unique tumor was used for analysis when duplicate or triplicate cores were provided for a single tumor. An additional TMA included a total of 43 normal pediatric tissues with duplicate cores for each tissue, other than placenta which included 7 cores, were stained for ALK with D5F3. Analysis for ALK staining in normal tissue included duplicate pituitary and testis samples for each age groups 0–2 years, 2–8 years, 8–12.5 years, and 12.5–18 years.

## Synthesis of CDX0239-PBD

The murine anti-ALK antibody was selected from our panel of generated ALK-directed monoclonal antibodies via immunization of mice with purified human ALK antigen encompassing the extracellular domain[29]. The purified ALK antigen encompassing the glycine-rich region and epidermal growth factor domain (T637-S1038) was expressed as a His-tagged protein in 293-F cells, purified by nickel affinity chromatography, and buffer-exchanged to phosphate-buffered saline (PBS) as previously described[29], and variable regions were humanized using homology modeling of the DNA sequence to match human IgG1, yielding 12 heavy and 8 light chains, and 96 combinations. These variants were expressed as human IgG1 and their affinity for recombinant human ALK was determined by bio-layer interferometry as previously described[38,39] to select the candidate humanized antibody (CDX0239) for ADC synthesis. The chimeric version of CDX0239 (CDX0125) was shown to protect a peptide in the C-terminal loop of the tumor necrosis-alpha (TNF-α)-like region of ALK (Y966-V972) which overlaps with activating peptide (ALKAL) binding site in hydrogen/deuterium exchange mass spectrometry (HDX-MS), indicating that CDX0239 binds to the extracellular domain proximal to the ALK cleavage site (N654-L655)[77]. Plasmids encoding the humanized anti-ALK IgG1 antibody (CDX0239) and its variant CDX0239-N297A harboring a mutation of the glycosylated asparagine at position 297 (N297) to alanine, were expressed in Expi293 cells (ThermoFisher). The antibodies were purified from cell culture medium using a column packed with Protein A resin (ThermoFisher). Purified antibodies were eluted using 50 mM acetic acid in Tris solution with pH of 9 to achieve

a final pH of 7.4. Eluted antibodies were dialyzed into 1x DPBS and concentrated using 10 kDa Ultra centrifugal filters (Amicon #UFC901008). The above protocol for antibody purification was used for all intermediate steps of ADC synthesis.

Synthesis of CDX0239-PBD from CDX0239 was completed in two steps. First, a three-branched linker (L6-linker, Amino-PEG4-Tris-PEG3-Azide, ConjuProbe #CP-2201) was conjugated to glutamine 295 (Q295) using microbial transglutaminase (mTG, Zedira #T250) which recognizes Q295 at the motif PWEEQYNST from the IgG1 heavy chain and attaches substrates baring primary amines[78]. As Q295 in IgG1 antibodies is typically sterically occluded by glycosylation of asparagine 297 (N297), CDX0239 was digested with peptide-N-glycosidase F (PNGase F, New England Biolabs #P0704S), which converted the glycosylated N297 to aspartic acid 297 (D297)[79]. Alternatively, the N297A mutation of CDX0239 as described above ensured the accessibility of Q295 by mTG without additional treatment. The three branched L6-linker was used at 5x excess concentration relative to the concentration of the Q295 sites and 1 U of mTG per 1 mg of CDX0239 was used and the reaction was incubated at 37 °C overnight. Second, the antibodies with conjugated L6-linker were purified and used in a click reaction for conjugation of the payload of PBD-dimer with a VA cleavage site, a PEG4 linker, and a dibenzocyclooctyne group (DBCO-PEG4-VA-PBD, Levena Biopharma #SET0306). The payload was used in excess from 1.5x to 1.9x the available azide sites and the reaction was incubated at 4 °C overnight. The quality of all reactions was monitored using liquid chromatography/mass spectrometry (LC/MS) with SQD2 mass-detector (Waters), which performed well for masses below 100 kDa and enabled measurement of denatured antibodies. All reactions except the click reaction for payload conjugation were extended to completion and produced single species. The click reaction for payload conjugation produced several species, where the species with two and three payloads per chain were dominant corresponding to DARs from 4 to 6. Attempts to increase the payload excess above 1.9x available azide site resulted in the aggregation and loss of the ADCs. We selected our candidate CDX0239-PBD with a DAR of 6 for further investigation after demonstrating superior in vitro activity in an ALK surface expression-dependent manner when compared to all other candidates.

## Neuroblastoma cell line culture

Neuroblastoma human cell lines NB-1 (derived from a metastatic lymph node of a 27 month old male), NB-SD (derived from a metastatic bone marrow lesion with age and sex of the patient unknown), IMR-32 (derived from an abdominal mass biopsy of a 13 month old female), NGP (derived from an unspecified site of a 30 month old male), and SK-N-AS (derived from a metastatic bone marrow lesion of an 8 year old female) were obtained from the Children's Hospital of Philadelphia Cell Bank and cultured in T75 tissue culture-treated flasks (Corning) with RPMI (Gibco) with 10% FBS (VWR), 1% L-glutamine (Gibco), and 1% penicillin-streptomycin (Gibco) and incubated at 37 °C 5% CO₂. Cells were passaged using Versine (Gibco) for 3 min at 37 °C 5% CO₂ before mechanical dissociation and resuspension in cell media. Short tandem repeat (STR) testing was completed on each cell line for authentication and routinely tested for *Mycoplasma* using MycoAlert.

## Generation of ALK knock-out NGP cell lines

Guides targeting *ALK* (5′-CCATACCTTAAATACGTAGG-3′, 5′-CTCTATTGCAGTTAGCGGAG-3′, 5′-CTGTAGCACTTTCAGAAGCG-3′, 5′-TCCAGACAACCCATTTCGAG-3′) were cloned into lentiCRISPR v2 (Addgene, #98290). Viruses were packaged by transfection of 293 T cells (ATCC) with packaging plasmids, pRSV-rev, pMD2 and pMDLg, and above guides or empty vector, utilizing Lipofectamine 2000 (Thermo Fisher Scientific #11668-019) according to manufacturer's instructions. *ALK* WT NGP cells were then transduced with produced lentiviruses and selected with puromycin at 5 μg/mL.

Selected cells were then seeded in 96 well plates for clone selection and individual clones were allowed to expand before harvesting. Knockout was confirmed via flow cytometry and western blot as per included protocols, and a successful clone was used as the NGP *ALK* knockout cell line.

### Determination of ALK surface expression in human and mouse tumors

Untreated parental tumors were collected after reaching volumes of 0.5-1.0 cm³ and were put into pre-warmed RPMI media (Gibco) and manually dissected with a scalpel. Human tumors were digested with enzymes R, H, and A within the tumor dissociation kit (Miltenyi Biotec #130-095-929) and mouse tumors were digested with collagenase (Type 4, ThermoFisher #17104019) and Dispase II (Sigma Aldrich #D4693), using a GentleMACS™ Dissociator (Miltenyi Biotec) at 37 °C according to the manufacturer's protocol. DNAse (Sigma-Aldrich #10104159001) was added to the mouse tumors and incubated at 37 °C for an additional 20 min. Samples were passed sequentially through 70 μm and 100 μm cell strainers for human tumors and through 40 and 70 μm strainer for mouse tumors. Red blood cells were depleted using the Red Blood Cell Lysis Solution (Miltenyi Biotec #130094183). For human tumors, mouse cells were depleted via cell labeling with a Mouse Cell Depletion Cocktail (Miltenyi Biotec #130104694) followed by magnetic separation with LS columns (Miltenyi Biotec #130042401) according to the manufacturer's protocol. Single cell suspensions were then washed with cold PBS and incubated with LIVE/DEAD™ Fixable Violet Dead Cell Stain (1:1000; Invitrogen #L23105) for 30 min. Cells were washed 3 times with cold PBS and incubated for 1 hour with CDX0239 anti-ALK primary antibody (50 nM) or IgG (50 nM), washed 3x with PBS, and incubated with goat anti-human IgG PE secondary antibody (1:200, Thermo Fisher Scientific #H10104) for 30 min. Cells were then washed 3x with cold PBS and fixed in 1% formaldehyde for 10 min. Stained cells were run on a Beckman Colter CytoFLEX S and analyzed using FlowJo software (v10.9.0). Determination of surface ALK expression in the NGP cell lines was completed following the above protocol for washing, staining, fixation, and analysis after collection from tissue culture.

### Internalization and lysosome localization of CDX0239 in NB cell lines

The internalization kinetics of CDX0239 in ALK-expressing neuroblastoma cells was determined using the Incucyte® Human Fabfluor-pH Orange Antibody Labeling Dye (Sartorius #4812). Briefly, cells were cultured at 10,000 cells/well in 50 μL of cell line specific-media in 96-well clear bottom tissue culture plates (Griener Bio-One) and incubated at 37 °C overnight. CDX0239 and IgG (Southern Biotech) were then conjugated with Incucyte® Human Fabfluor-pH Orange Antibody Labeling Dye per the manufacturer's protocol and 50 μL of conjugated corresponding antibody was added to appropriate cells and analyzed with 4x orange phase images per well every hour for 48 h, and antibody internalization was determined using the orange integrated intensity per image measurement on the Incucyte® Zoom software (Sartorius). Dual labeling with LysoSensor Green (Thermo Fisher Scientific #L7535), a lysosomal marker, was completed in tandem with CDX0239 conjugated to Incucyte® Human Fabfluor-pH Red Antibody Labeling Dye (Satorius #4722). Briefly, 100,000 NB-1 or 50,000 SK-N-AS cells were seeded on cover slips (MatTek corporation) coated with retronectin (Takara Bio) and allowed to adhere overnight. CDX0239 conjugated to Incucyte Human Fabfluor-pH Red Antibody Labeling Dye (4μg/ml) was added and 3 h later, LysoSensor Green (0.5μM) was added. Fluorescence images were captured using a custom-built lattice light-sheet microscope (LLSM)[80]. Specifically, we conducted simultaneous dual-color, lattice light sheet imaging using the 488 nm wavelength laser at 0.04 mW power to image the LysoSensor Green and the 642 nm wavelength laser at 1.8 mW to image the Fabflour Red. The

exposure time was set to 50 ms for all the images. Laser powers and exposure time were all kept consistent across the different conditions. The images were analyzed using ImageJ software.

### CDX0239-PBD-Induced Cell Line Cytotoxicity and Caspase Activity

ADC cytotoxicity and caspase induction in neuroblastoma cell lines were determined using CellTiter-Glo® 2.0 cell viability assay (Promega #G9242) and Caspase-Glo® 3/7 assay (Promega #G8092), respectively. Neuroblastoma cell lines were cultured in 96-well clear bottom tissue culture plates (Greiner Bio-One) at 5000 cells/well in 90 μL of cell culture media as above and incubated at 37 °C overnight. Neuroblastoma cell lines were then treated with 10 μL of either CDX0239-PBD, CDX0239 without PBD payload, free PBD (The Chemistry Resource Solution), IgG (Southern Biotech), or cell culture media at concentrations from $1 \times 10^{-15}$ to $1 \times 10^{-8}$ M and incubated at 37 °C for 5 days. After treatment, plates were acclimated to room temperature for 30 min then treated with either CellTiter-Glo® 2.0 or Caspase-Glo® 3/7 per the manufacturer's protocol and analyzed on the Glomax® Discover (Promega).

### Neuroblastoma cell line competition assay

CDX0239-PBD ALK site-specific binding and subsequent internalization-dependent cytotoxicity was determined using a competition assay. Briefly, NB-1 neuroblastoma cells were plated in 96-well clear bottom tissue culture plates (Griener Bio-One) at 5000 cells/well in 80 μL of cell culture media stated above and incubated at 37 °C overnight. NB-1 neuroblastoma cells were treated with 10 μL of media containing CDX0239-PBD to produce a final culture concentration of 8.0 pM in addition to 10 μL of media containing CDX0239 without payload with increasing concentrations from 0–1.2 μM. NB-1 neuroblastoma cells were then cultured for 5 days and then analyzed for cell viability using the CellTiter-Glo® assay described above and assessed for viability in reference to control NB-1 cells treated with media only.

### In vivo CDX0239-PBD studies

Female CB17 SCID mice (Charles River) were used for animal studies and housed in female mice-only Supermouse 750 Micro-Isolator cages at 5 mice per cage with corncob bedding, with light cycles starting at 0600 and dark cycles starting at 1800. Temperature was maintained between 20–26 °C and humidity was maintained between 30–70%. Female mice were used for all animal studies to ensure appropriate animal housing conditions. Cell line-derived xenografts (CDXs; NB-1, NB-SD, SK-N-AS, SW-48, and HCT-116) or patient-derived xenografts (PDXs; COG-N-424x and RH-41) were implanted subcutaneously into the right flank of female CB17 SCID mice at an age of 6–8 weeks. When tumors reached 0.20–0.25 cm³, the animals were randomized into groups of 5 and dosed intraperitoneally for three consecutive weeks for three total weekly injections of 1 mg/kg CDX0239-PBD, IgG, CDX0239, CDX0239 with deglycosylation via asparagine to alanine mutation, CDX0239 with deglycosylation via enzymatic digestion, CDX0239 with deglycosylation via asparagine to alanine mutation and with linker, or normal saline vehicle (10 mL/kg). A separate study included xenograft NB-1 models treated with a single dose of 1 mg/kg CDX0239-PBD compared to treatment with normal saline vehicle (10 mL/kg). A separate study included xenograft NB-SD models treated with daily 10 mg/kg lorlatinib via gavage compared normal saline vehicle (10 mL/kg). Tumor volumes were determined using a spheroid formula ($V = ((L/2 + W/2)^3 * p/6)/1000$, where V = volume in cm³, L = length in cm, and W = width in cm) and total body weights were recorded a minimum of twice a week. Mice were euthanized via carbon dioxide inhalation from a compressed gas cylinder when tumor size was at the maximal permitted size of ≥2 cm³, and this maximal tumor size was not exceeded when a single tumor measurement was found to be ≥2 cm³. Death was ensured by cervical dislocation after cessation of

breathing as a secondary method of euthanasia. Mouse samples analyzed for ALK surface expression were collected from Th-MYCN mouse models as previously described[40] following euthanasia as above and sectioned into 5 mm segments for processing for flow cytometry as stated above.

## Western blot analysis

Confirmation of the cytotoxic mechanism of CDX0239-PBD was completed using Western Blot analysis. NB-SD xenograft tumors were collected 3 and 7 days after treatment with normal saline vehicle (10 mL/kg) or CDX0239-PBD (1 mg/kg) and processed using 200–500 μL of hypotonic lysis buffer (10 mM NaCl, 10 mM Tris HCl, 1 mM ethylenediaminetetraacetic acid (EDTA), 1% Triton X-100) with 1x protease/phosphatase inhibitor (Cell Signaling, #58727) and homogenized with TissueRuptor II (Qiagen, #9002755) with TissueRuptor Probes (Qiagen, #990890) on ice. Lysates were separated at 18,000 xG for 15 min at 4 °C and the supernatant containing protein lysate was transferred to new Eppendorf tubes. Protein content was quantified using a BCA quantification kit (ThermoFisher Scientific #23225) per the manufacturer's protocol. Samples were loaded to gels at 40 μg/well, resolved at 120 V for approximately 2 h, and transferred to 0.45 μM PVDF membranes. The membranes were blocked for 1 hour at room temperature with 5% BSA or 5% milk to match primary antibody blocking dilution buffer in tris-buffered saline with polysorbate 20 (TBST, Cell Signaling #9997) and stained with the following primary antibodies in 5% BSA or 5% milk in TBST to match blocking buffer at a dilution of 1:1,000: total ALK (Cell Signaling, #3633S, Lot #13), phosphorylated ALK (pALK Y1604, Cell Signaling, #3341 L, Lot#7), γH2AX (Cell Signaling, #2577S, Lot #14), cleaved caspase-3 (Cell Signaling, #9664 L, Lot #22), PARP (Cell Signaling, #9532S, Lot #10), and β-actin (Cell Signaling, #4967S, Lot #10) overnight on a shaker at 4 °C. All primary antibodies were sourced from rabbit. Membranes were then washed 3x for 10 min with TBST at room temperature, stained with appropriate goat anti-rabbit HRP-conjugated secondary antibody (Cell Signaling) in 5% BSA or 5% milk in TBST to match blocking buffer at a dilution of 1:5,000, incubated with Amersham™ ECL™ Western Blotting Detection Reagents (Cytiva #RPN2232), and analyzed on an Azure Biosystems Sapphire Biomolecular Imager. Uncropped and unprocessed scans are included within the Source Data file.

## Statistical & reproducibility

All in vitro assays were conducted with at least three technical replicates per condition. All in vivo studies were conducted with an $n = 5$ mice per treatment arm to achieve adequate statical power for analysis. ALK RNA expression data was generated by polyA selection method and quantifies as log2(TPM) as above. A two-way ANOVA was used to compare mean values between multiple groups. A three-parameter log vs response analysis was used to determine $IC_{50}$ values. Square root transformations were performed on tumor volumes to normalize distribution, and a linear mixed effects model was constructed to estimate differences in the rate of tumor volume changing over time between different groups. This model included group, day, and group-by-day interactions as fixed effects, with a significant group-by-day interaction suggesting different rates of tumor volume changes between groups. This model used the vehicle-treated group as the reference group and were compared to treatment groups using a linear mixed-effects p value. Survival curve analysis was completed using the Kaplan-Meier method. All data was analyzed through GraphPad Prism version 8 or later. A $p < 0.05$ was considered statistically significant and all in vitro experiments were reproducible. In vitro experiments were not randomized, and all in vivo experiments were randomized. The investigators were not blinded to allocation during in vivo experiments and outcome assessment. No data were excluded from the analyses.

## Reporting summary

Further information on research design is available in the Nature Portfolio Reporting Summary linked to this article.

## Data availability

Source data are provided with this paper. The datasets generated during and/or analyzed during the current study are available in the main text, supplementary information, or source data provided with this manuscript. Individual histology slides and pictographs used in this study are available from the corresponding author upon request. All biologic material other than CDX0239 and CDX0239-PBD is available upon request due to intellectual property restrictions. Cell lines and xenografts may be made available upon request pending resource restriction. The links for the publicly available datasets from the Tumor Compendium v11 of UCSC Treehouse Childhood Cancer Initiative and from The Genotype-Tissue Expression Project v8 used for ALK RNA expression analysis to generate Fig. 1, Supplemental Table 1, and Supplemental Table 2 are provided below. UCSC Treehouse Childhood Cancer Initiative Tumor Compendium v11: The Tumor Compendium v11 is available from the UCSC Treehouse Childhood Cancer Initiative portal at https://treehousegenomics.soe.ucsc.edu/public-data/. Direct download and access instructions are provided on the portal. No restrictions apply. The Genotype-Tissue Expression (GTEx) Project v8: The GTEx data set is available from the GTEx portal at http://gtexportal.org/home/datasets. All processed RNA sequencing data are publicly available and raw data may require dbGap authorization (Accession: phs000424.v8.p2). Source data are provided with this paper.

## Code availability

ALK RNA expression visualization as boxplots were generated in R version 4.2.3 using ggplot2 function from Tidyverse package version 2.0.0. Code scripts are available at the Github repository marislab/ALK_ADC v1.0.1 and is archived via Zenodo (https://doi.org/10.5281/zenodo.15830580)[81]. The repository includes the code used to determine histologies that were outliers of ALK RNA expression to generate Fig. 1 A-B in this manuscript.

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

## Acknowledgements

This work was supported in part by the National Cancer Institute grant R01CA140198-11-15 (Y.P.M.), Patricia Brophy Endowed Chair in Neuroblastoma Research (Y.P.M), DOD Award W81XWH-12-1-0486 (Y.P.M.), the National Institutes of Health grant R013208130624 (Y.P.M), the National Institutes of Health grant DP2HD108775 (Y.P.M), funding from Braden's Hope Foundation (Y.P.M.), the Margaret Q Landenberger Foundation, the Howard Hughes Medical Institute (HHMI), National Cancer Institute grant R37CA282041 (K.R.B), and National Cancer Institute grant K08CA230223 (K.R.B.). Author A.D.G. is supported by NIH Grant 2T32CA009615 Author M.M. is supported as a HHMI Freeman Hrabowski Scholar.

## Author contributions

A.D.G. Conceptualization, methodology, validation, formal analysis, investigation, writing – original draft, visualization, project administration S.M. Conceptualization, methodology, validation, formal analysis, investigation, writing – review & editing, visualization C.A. (Christina Acholla) Methodology, validation, formal analysis, investigation, visualization, writing – review & editing C.C. (Colleen Casey) Investigation, writing – review & editing G.L. Investigation, writing – review & editing, visualization M.M. (Martina Mazzeschi) Investigation K.P. Investigation, formal analysis, data curation K.K. Visualization, writing – review & editing C.C. (Chuan Chen) Methodology, resources S.B. Investigation, writing – review & editing J.K. Investigation, visualization, writing – review & editing P.K. Investigation, writing – review & editing M.G. Methodology, investigation, writing – review & editing G.P. Investigation, writing – review & editing D.G. Methodology, investigation A.M. Investigation, formal analysis C.A. (Cynthia Adams) Methodology, resources G.W. Methodology, investigation A.K.H. Methodology, investigation D.M. Formal analysis, investigation, resources J.P. Formal analysis, investigation, resources T.T.S. Investigation Y.L. Formal analysis, writing – review & editing P.J. Investigation, visualization D.A. Methodology, resources M.L. Investigation J.M.M. Resources, writing – review & editing, supervision A.J.W. Resources, writing – review & editing K.R.B. Investigation D.D. Methodology, resources M.M. (Mustafa Mir) Methodology, resources D.V.J. Methodology, resources Y.P.M. Conceptualization, methodology, resources, writing – original draft, visualization, supervision, project administration, funding acquisition.

## Competing interests

The authors declare no competing interests.

## Additional information

**Supplementary information** The online version contains
supplementary material available at

Yael P. Mossé.

Alberto D. Guerra [1], Smita Matkar[1], Christina Acholla[1], Colleen Casey [1], Grant Li[1], Martina Mazzeschi [2], Khushbu Patel[1], Kateryna Krytska[1], Chuan Chen[3], Skye Balyasny [1], Joshua Kalna [1], Paul Kamitsuka[1], Mark Gerelus [1], Grace Polkosnik [1], David Groff [1], Apratim Mukherje[4], Cynthia Adams[3], Gabriela Witek[1], Amber K. Hamilton[1], Daniel Martinez[1], Jennifer Pogoriler [1], Timothy T. Spear[1], Yimei Li[1], Piotr Jung[1], Diego Alvarado[5], Mattia Lauriola [2,6], John M. Maris [1,7], Adam J. Wolpaw [1,7], Kristopher R. Bosse [1,7], Dimiter Dimitrov[3], Mustafa Mir [8], Dontcho V. Jelev[3] & Yael P. Mossé [1,7] ✉

[1]The Children's Hospital of Philadelphia, Division of Oncology and Center for Childhood Cancer Research, Philadelphia, PA, USA. [2]IRCCS Azienda Ospedaliero-Universitaria di Bologna, Bologna, Italy. [3]University of Pittsburgh Medical School, Department of Medicine, Division of Infectious Diseases, Center for Antibody Therapeutics, Pittsburgh, PA, USA. [4]University of Pennsylvania, Department of Cell and Developmental Biology, Philadelphia, PA, USA. [5]Celldex Therapeutics, Antibody Therapeutics Department, Hampton, NJ, USA. [6]University of Bologna, Department of Medical and Surgical Sciences, Bologna, Italy. [7]Perelman School of Medicine at the University of Pennsylvania, Department of Pediatrics, Philadelphia, PA, USA. [8]The Children's Hospital of Philadelphia, Center for Computational and Genomic Medicine, Philadelphia, PA, USA. ✉e-mail: mosse@chop.edu

