## [Transparent Peer Review file · Nature Communications]

A humanized anaplastic lymphoma kinase (ALK)-directed antibody-drug conjugate with pyrrolobenzodiazepine payload demonstrates efficacy in ALK-expressing cancers

Corresponding Author: Professor Yael (P.) Mossé

Version 0:

Reviewer comments:

Reviewer #1

(Remarks to the Author)

Yael Mosse ALK ADC Nat comms comments

Figure 2; Immunohistochemistry uses antibody clone D5F3 but neither text nor methods informs the reader which epitope this antibody binds and if the sections are being scored for total expression or membrane staining. In a quick search of literature I could not find information on the position of the epitope. This is of particular relevance given the known cleavage of ALK in neuroblastoma. If the D5F3 epitope were N terminal to the cleavage site then the staining intensity of neuroblastoma is likely to be under-represented.

Figure 2C-D; staining in pituitary is described as equivocal but it is hard for the reader to appreciate what this means. Does this mean some rare cells with bright membrane staining or all cells with faint and or indeterminate staining which could be for example cross reactivity with a cytoplasmic antigen. There is not enough detail in results or methods section to determine this and the resolution of figure 2D, at least in the pdf provided to reviewers, is too low to appreciate the staining.

Figure 3. There is inadequate information to assess the cell staining. The samples are described as xenografts but there is no section in the methods on in vivo tumour growth; is it subcutaneous or orthotopic? What size of tumour? – these factors can influence the amount of stroma that needs to be gated out. What gating was used to gate out myeloid and endothelial cells, doublets etc? Without the gating information can we exclude ALK staining on stroma? Most importantly there is no information on CDX0239 epitope. Is it membrane proximal or distal to the ALK cleavage site? That is of utmost importance since if CDX0239 is membrane distal to the cleavage then cells with cleaved ALK may become treatment escape variants. The data in figure 3 is also of great importance to evaluate translational significance of subsequent in vivo studies. The NB-1 tumour model shows what looks like 3 log higher MFI than the isotype control. This appears to be very significantly higher than typical flow plots of neuroblastoma using other antibodies (for example Walker et al PMID 28676342). Of course this might be because of relative brightness of the respective fluorophores used. Hence, it would be of great value to provide objective antigen expression data for the targets used in the paper using for example Quantibright beads.

Figures 5-7. The efficacy of the CDX0239-PBD ADC is shown in figures 5-7 with in vitro in 5 and in vivo in 6+7. In figure 5 the important control of NGP targets versus NGP knock-out is shown; this convincingly shows that in vitro cytotoxicity is ablated once ALK surface expression by genetic ablation, which would be strongly suggestive that the ADC activity is dependent on specific antibody to antigen interaction. In 5C, the PBD sensitivity is shown in molar values and appears to show equivalent sensitivity of NGP cells towards free PBD versus CDX0239-PBD. However, the other target lines NB-1 and ND-SD show higher sensitivity to the ADC than to free PBD and this maybe correlates with the somewhat higher expression of ALK in NB-1 and NB-SD than in NGP. The data suggests that targets such as SK-N-AS have a subthreshold ALK expression resulting in resistance to the ADC whilst NB1 and NB-SD are suprathreshold and are sensitive. NGP is more complex because of the equivalent IC50 to free payload and ADC; this maybe is related to intermediate ALK expression. The concern this reviewer has is whether NB-1 and NB-SD therefore represent outliers amongst the neuroblastoma cases,

whilst SK-N-AS and NGP might be more representative of typical levels of ALK expression. These concerns could be allayed in several ways: 1) the quantification of ALK expression in the targets used in the paper in terms of molecules per cell; 2) in vivo data from NGP-WT versus NGP-KO (was this done; if yes then including it would strengthen the paper even if the ADC has no significant benefit versus control); 3) inclusion of any additional animal studies in ALK positive targets where the ADC is ineffective; or a statement that there are not any; 4) mechanistic insight as to why SK-N-AS is resistant to the ADC despite showing ALK expression- it might be failure of internalisation in SK-N-AS related to avidity caused by low antigen expression compared to the other targets as shown in figure 4. Are there internalisation data on targets such as COG-N-424x or HCT-116 which looks to be similarly antigen dim but have high sensitivity and good in vivo efficacy. Antibody controls with modifications to reduce efficacy are included in supplemental figures 3 and 4 but the text (results or figure legends) does not explain what the controls represent nor discuss relevance. Overall this reviewer is excited to see the data especially given the additional strong responses in the HCT, SW48 and RH41 models- but addressing the concerns above would strengthen the conclusion that the mechanism is ALK and internalisation dependent. Is there a simple threshold of expression for efficacy (which could imply a future predictive biomarker) or are there other factors at play related to ALK internalisation kinetics.

Figure 8 shows the NB-SD model to be sensitive to ADC but resistant to lorlatinib. It would be important to know if the data in 8B and 8D is the same as 5A or an experimental repeat. They look very similar and so if a repeat, this would add to the story and should be made clear in the narrative.

Reviewer #2

(Remarks to the Author)

The manuscript is well-structured, and the experiments are generally well designed and executed. However, there are a few aspects that require further clarification and additional data to strengthen the conclusions, particularly regarding the therapeutic potential of CDX0239-PBD in CNS malignancies.

Major Comments

1. IC50 Determination: The authors should provide the IC50 value of CDX0239-PBD to allow for a better understanding of its potency.
2. Blood-Brain Barrier Considerations: Given that CDX0239-PBD would need to cross the blood-brain barrier to be effective against CNS malignancies, the expected concentration of the drug in the brain should be discussed. Experimental data or relevant pharmacokinetic modeling would strengthen this aspect.
3. ALK Expression Assessment: The authors use ALK-D5F3 for immunohistochemical staining in FFPE TMA slides. The staining in Figure 1 appears to be strong, with a moderately high H-score. How does this compare to ALK-amplified or overexpressed tumors? Providing a comparative analysis would enhance the interpretation of these results.
4. Expression Threshold for Efficacy: At what expression level (e.g., copy number threshold) does CDX0239-PBD lose efficacy? This information is critical for understanding patient selection criteria.
5. Drug-Antibody Ratio (DAR) and Pharmacokinetics: The manuscript states a DAR of 6 for CDX0239-PBD. The authors should provide the in vivo pharmacokinetic (PK) profile, including measurements of free PBD levels, to evaluate potential systemic toxicity and drug clearance.
6. In Vivo Study Design: The efficacy argument for CDX0239-PBD could be further strengthened by including an additional treatment arm with PBD alone at equimolar concentrations. This would help distinguish the contribution of the ADC from that of the free payload.

Minor Comments

- Typographical Error: Line 332 contains a minor typo where "am" should be corrected to "an."

Reviewer #3

(Remarks to the Author)

In this manuscript, A. Guerra et al. describe a preclinical evaluation of CDX0239-PBD, a novel antibody-drug conjugate (ADC) targeting Anaplastic Lymphoma Kinase (ALK), a lineage-restricted antigen overexpressed in neuroblastoma (NB) and several other pediatric and adult malignancies. Utilizing a humanized ALK-specific monoclonal antibody conjugated to a pyrrolobenzodiazepine (PBD) payload, the study demonstrates potent and selective cytotoxicity in ALK-expressing tumor cells, with minimal off-target effects in normal tissues. The authors provide extensive in vitro and in vivo evidence of ALK surface expression-dependent internalization, lysosomal trafficking, and DNA-damage-mediated apoptosis, leading to sustained tumor regression in multiple xenograft models. These findings suggest that CDX0239-PBD is a promising candidate as a precision therapeutic for ALK-expressing cancers.

Overall, the study represents a significant advance in targeted therapies for NB and other ALK-positive malignancies. The experimental design is rigorous, with appropriate positive and negative controls, in vitro and in vivo validation, and the use of multiple ALK-expressing cancer models. Notably, demonstrated minimal ALK expression in normal tissues decreases the risk of CDX0239-PBD on-target toxicity. Mechanistically, the results demonstrate that the ADC works via ALK internalization and lysosomal localization leading to induction of apoptosis in cancer cells. However, there are the following concerns:

- 1) The reliance on transplantable xenograft models limits the understanding of CDX0239-PBD efficacy against spontaneous tumors with their complex TME. Authors could try a syngeneic NB model or at least acknowledge the limitations of xenograft models in discussion.
- 2) Figure 2 lacks an analysis of ALK surface expression's association with MYCN amplification and patient age, independent prognostic markers in NB.
- 3) Although the study discusses ADCs' ability to overcome TME-related barriers, more in-depth exploration of how CDX0239-PBD penetrates tumor vs. normal tissues (e.g. liver) would strengthen the translational potential.

Reviewer #4

(Remarks to the Author)

The submitted manuscript, "A humanized ALK-directed antibody-drug conjugate CDX0239-PDB demonstrates potent efficacy in ALK-expressing cancers" by Guerra and colleagues, presents a compelling approach to precision chemotherapy development. Over more than a decade, the authors have advanced this research toward a potential clinical application. Beginning with a murine antibody capable of blocking ALK activity, the researchers have systematically improved its properties through modifications, enhancing its affinity and specificity. Ultimately, they generated a humanized ALK-directed monoclonal antibody with high affinity, conjugated to pyrrolbenzodiazepine. The manuscript presents promising results, demonstrating significant responses in ALK-expressing cell line xenografts and opening a viable path for clinical development.

Comments

- Lines 95–96: It seems odd that reference 2 appears both in the middle and at the end of the sentence. The authors are making a point, but scientists in the field are already aware of these facts.
- Lines 104–106: The reference is incorrect. Where has this manuscript been published? The authors should also include additional references, as this observation has been reported elsewhere, such as in PMID: 20719933, PMID: 26121087, and PMID: 25071110.
- Line 110: The authors should add another reference to emphasize that most patients harboring neuroblastoma tumors express ALK (PMID: 38229432).
- Lines 120–125: This sentence could be improved for clarity, making it more accessible to general readers. Additionally, several long sentences throughout the manuscript should be refined for readability.
- Line 161: Add reference PMID: 38229432, which documents ALK expression in neuroblastoma patients. This should be included, as the authors validate previously published findings effectively.
- Line 184: While the methods section clarifies this to some extent, the authors should provide more details on the methodology and procedure for generating CDX0239. Was CDX0239 derived from the discovery of CDX125, as referenced in reference 31, or from PMID: 30867324? Additionally, where was the murine anti-ALK antibody originally obtained?
- Line 185: The authors use the bio-layer interferometry method. It would be appropriate to cite a more foundational article describing this technique, such as PMID: 22414486.
- Lines 185–203: Is CDX0239 activating or inhibiting ALK activity?
- Line 313: CDX0239-PBD did not affect phosphorylated ALK (pALK1604) levels. What about phosphorylated pALK1278, which is necessary for ALK activity? The authors should verify their data using a pALK1278 antibody. Have they observed any additional differences in downstream ALK signaling when using CDX0239? This is not addressed in the manuscript.
- Line 398: The authors should include reference PMID: 34646012, as it was published in the same issue as reference 59.
- Line 453: The specificity and selection process for the humanized antibody (CDX0239) are unclear. Which ALK epitope does it bind to? Is it ALK or ALK/ALKAL2? The authors refer to reference 31, where they express a codon-optimized cDNA encoding the extracellular glycine-rich domain of ALK (aa 678-1030) fused to the ALK ligand ALKAL2 (aa 71-152). Could it be that the antibody only recognizes ALK RTK when bound to an ALK ligand? The authors should clarify this.

Version 1:

Reviewer comments:

Reviewer #1

(Remarks to the Author)

Thank you to the authors for detailed amendments to address all the issues and questions I had raised (reviewer 1). I consider this to be important work of value to the field and of sufficient scientific rigor for publication.

Reviewer #2

(Remarks to the Author)

The authors have addressed the majority of my original comments with additional clarification and relevant supporting data. Their responses—particularly regarding pharmacologic activity, therapeutic relevance in CNS malignancies, and ALK expression thresholds—have improved the overall clarity and scientific completeness of the manuscript. While some areas—such as pharmacokinetics and alternative approaches to measuring clinically relevant ALK expression—could benefit from further refinement in future work, the revisions are sufficient to support publication in the current form.

Reviewer #3

(Remarks to the Author)

The authors have satisfactorily addressed my critiques.

Reviewer #4

(Remarks to the Author)

The revised manuscript, "A humanized ALK-directed antibody-drug conjugate CDX0239-PDB demonstrates potent efficacy in ALK-expressing cancers" by Guerra and colleagues, has been improved. The authors have clarified the manuscript and readability on all points observed by this reviewer. The authors should be proud that they advanced this research toward a potential clinical application, which is needed for full length ALK-driven cancers

REVIEWER 1

Comment 1. *Figure 2; Immunohistochemistry uses antibody clone D5F3 but neither text nor methods informs the reader which epitope this antibody binds and if the sections are being scored for total expression or membrane staining. In a quick search of literature I could not find information on the position of the epitope. This is of particular relevance given the known cleavage of ALK in neuroblastoma. If the D5F3 epitope were N terminal to the cleavage site then the staining intensity of neuroblastoma is likely to be under-represented.*

Response: We have now provided additional information about antibody clone D5F3 in the manuscript (**lines 172-173 and lines 525-527**). Rabbit anti-ALK D5F3 (Cell Signaling, #3633) is an FDA and Clinical Laboratory Improvement Amendments (CLIA)-approved antibody as a standalone test for ALK rearrangements in lung cancer. While the binding epitope is not available in the literature as it is proprietary, it is known that D5F3 binds to the cytoplasmic kinase domain of ALK and hence is able to recognize ALK fusion proteins which are lacking the extracellular, N-terminus domain of ALK (**Sung and Kim. Thoracic Cancer. 2024. PMID: 39257160**). As such, this antibody provides interpretable and broadly relevant staining intensity of total full length ALK in our neuroblastoma tissue microarray (TMA) samples.

Comment 2. *Figure 2C-D; staining in pituitary is described as equivocal but it is hard for the reader to appreciate what this means. Does this mean some rare cells with bright membrane staining or all cells with faint and or indeterminate staining which could be for example cross reactivity with a cytoplasmic antigen. There is not enough detail in results or methods section to determine this and the resolution of figure 2D, at least in the pdf provided to reviewers, is too low to appreciate the staining.*

Response: We now define equivocal staining as “minimally present staining of a cell with anti-ALK D5F3 at 1:500 dilution” and we previously defined the calculation the total H-score for each tumor or tissue sample based on the percent of cells with the most prominent intensity (1 representing equivocal, 2 weak, and 3 strong positive staining; **lines 537-540**). We have updated Figure 2 to include a high-powered pictograph (20x magnification) of a diagnostic neuroblastoma tumor core from our patient TMA stained with anti-ALK D5F3 (**Figure 2C**) for qualitative comparison to our pituitary samples (**Figure 2E**) that demonstrated equivocal staining in a subset of cells, and this is now further clarified in the Results section (**lines 193-197**). The represented diagnostic neuroblastoma core example in **Figure 2C** is from a right neck mass diagnostic biopsy from a 9-year-old male with high-risk neuroblastoma (*MYCN* non-amplified and *ALK* wild-type). This core was selected for representation as it has an H-score of 142.33 which reflects the near average H-score of 144.71 for ALK staining in the neuroblastoma tumor TMA (**Figure 2C**).

Comment 3. *Figure 3. There is inadequate information to assess the cell staining. The samples are described as xenografts but there is no section in the methods on in vivo tumour growth; is it subcutaneous or orthotopic? What size of tumour? – these factors can influence the amount of stroma that needs to be gated out. What gating was used to gate out myeloid and endothelial cells, doublets etc? Without the gating information can we exclude ALK staining on stroma? Most importantly there is no information on CDX0239 epitope. Is it membrane proximal or distal to the ALK cleavage site? That is of utmost importance since if CDX0239 is membrane distal to the cleavage then cells with cleaved ALK may become treatment escape variants.*

Response: We agree that variables such as tumor size and the site of xenograft implantation can influence ALK surface expression and overall efficacy of ALK-directed therapies. We clarify that tumors were implanted subcutaneously into the right flank of female CB17 SCID mice and randomized into treatment groups when tumors reached 0.20-0.25 cm³ and treated via intraperitoneal injection weekly for three consecutive weeks at 1 mg/kg of defined treatment or 10 mL/kg of normal saline vehicle (**lines 679-685**). Untreated parental tumors were assessed for ALK surface expression after reaching volumes of 0.5-1.0 cm³, and we have now included details on xenograft tumor size before flow cytometry analysis for surface ALK expression (**line 613**). We employed SCID mice for our *in vivo* studies and utilized a mouse cell depletion kit (Miltenyi Biotec, #130-104-694) to remove mouse myeloid, endothelial, and stromal cells to eliminate any confounding background staining. Additionally, we have included an example of our gating strategy used for analysis of our flow cytometry data after mouse cell depletion (**Supplemental Figure 2**).

Regarding the CDX0239 epitope, the original ALK-directed mouse monoclonal antibody was generated against human ALK comprising the glycine rich and epidermal growth factor domain (T637-S1038; **Sano et al. Sci Transl Med. 2019. PMID: 30867324**). Although the crystal structure of the ALK extracellular domain bound to the antibody is not available, the chimeric version of the ALK-directed antibody (CDX0125) was shown to protect a peptide in the C-terminal loop of the TNF- α like region of ALK (Y966-V972) which overlaps with activating peptide (ALKAL) binding site in hydrogen/deuterium exchange mass spectrometry (HDX-MS; **Li et al. Nature. 2021. PMID: 34819665, Figure 4**), indicating that the antibody binds to the extracellular domain proximal to the ALK cleavage site (N654-L655; **Huang et al. Cell Rep. 2021. PMID: 34260934**). The antibody used for ADC synthesis (CDX0239) was humanized via homology remodeling of the DNA sequence for the variable regions to match human IgG1, yielding 12 heavy and 8 light chains, and 96 combinations. Plasmids encoding the humanized anti-ALK IgG1 antibody were cloned in Expi239 cells to produce the humanized anti-ALK monoclonal antibody and is now described in detail in the manuscript (**lines 549-564**). Importantly, for the determination of off-target binding and toxicity of CDX0239-PBD, we previously confirmed species cross-reactivity of CDX0239 with murine spontaneously occurring neuroblastoma tumors (**lines 203-207 and Supplemental Figure 1**).

Comment 4. *The data in figure 3 is also of great importance to evaluate translational significance of subsequent in vivo studies. The NB-1 tumour model shows what looks like 3 log higher MFI than the isotype control. This appears to be very significantly higher than typical flow plots of neuroblastoma using other antibodies (for example Walker et al PMID 28676342). Of course this might be because of relative brightness of the respective flourophores used. Hence, it would be of great value to provide objective antigen expression data for the targets used in the paper using for example Quantibright beads.*

Response: We agree with the reviewer that determination of ALK cell surface expression is of great importance to evaluate the translational potential of CDX0239-PBD. The NB-1 xenograft model used in our study harbors high-level *ALK*-amplification, with >60 copies of the *ALK* gene as previously demonstrated, which at the protein level is robustly overexpressed (**Bresler et al. Sci Transl Med. 2012. PMID: 22072639**). This model represents the 3-4% of patients with high-risk neuroblastoma whose tumors harbor *ALK*-amplification (**Berko et al. J Clin Oncol. 2025. PMID: 40036726**). The COG-N-424x, NB-SD, and NGP neuroblastoma xenograft models demonstrate moderate ALK cell surface expression (**Figure 3A**), which is much more representative of most neuroblastoma tumors. These xenograft models demonstrate geometric means for surface ALK expression on flow cytometry that are similar to other xenograft models of neuroblastoma (**Walker et al. Mol Ther. 2017. PMID: 28676342, Figure 1B**). Importantly, we utilized the same humanized ALK-directed monoclonal antibody (CDX0239) for flow cytometry analysis of ALK surface expression and for synthesis of CDX0239-PBD and demonstrate that our estimated ALK surface expression by ALK:IgG geometric mean is positively correlated with efficacy of CDX0239-PBD. This is now clarified in the Discussion section (**lines 400-408**).

We attempted to conjugate a fluorophore to CDX0239 for Quantibrite™ bead (BD Biosciences, #340495) analysis. Quantibrite™ beads are intended for estimation of antibodies bound per cell (ABC) with fluorophore-conjugated antibodies as the beads link directly to the conjugated fluorophore. We were unable to successfully conjugate a fluorophore to CDX0239 despite multiple attempts as conjugation of a fluorophore led to loss of CDX0239 affinity for surface ALK. The effect of decreased antibody affinity after conjugation of fluorophores has been previously demonstrated (**Szabó et al. Biophys J. 2018. PMID: 29414714**). A quantification protocol utilizing a QIFKIT assay (Agilent Dako, #K007811-8) would be an alternative approach for quantification, however this kit is only available for mouse monoclonal antibodies which would not be compatible with our humanized ALK-directed antibody CDX0239. Additionally, full-length IgG humanized antibodies such as CDX0239 have non-specific binding to human cells despite utilizing an Fc block and bovine serum albumin (BSA) block. The non-specific binding of our human IgG control varied substantially between xenograft models but was consistent within xenograft models and enabled us to calculate the ALK:IgG geometric mean of ALK surface expression in comparison to non-specific binding of the IgG control in each xenograft model. This resulted in an effective quantitative representation of ALK surface expression for each xenograft model. We now detail this in the Results section along with previously stated ALK:IgG geometric means (**lines 207-224 and Supplemental Table 5**).

Comment 5. *Figures 5-7. The efficacy of the CDX0239-PBD ADC is shown in figures 5-7 with in vitro in 5 and in vivo in 6+7. In figure 5 the important control of NGP targets versus NGP knock-out is shown; this convincingly shows that in vitro cytotoxicity is ablated once ALK surface expression by genetic ablation, which would be strongly suggestive that the ADC activity is dependent on specific antibody to antigen interaction. In 5C, the PBD sensitivity is shown in molar values and appears to show equivalent sensitivity of NGP cells towards free PBD versus CDX0239-PBD. However, the other target lines*

NB-1 and NB-SD show higher sensitivity to the ADC than to free PBD and this maybe correlates with the somewhat higher expression of ALK in NB-1 and NB-SD than in NGP. The data suggests that targets such as SK-N-AS have a subthreshold ALK expression resulting in resistance to the ADC whilst NB1 and NB-SD are suprathreshold and are sensitive. NGP is more complex because of the equivalent IC50 to free payload and ADC; this maybe is related to intermediate ALK expression.

Response: We agree that it is of immense importance to address the ALK cell surface expression-dependent therapeutic threshold in relation to anti-tumor activity of CDX0239-PBD. We have addressed these concerns in detail below:

(a) The concern this reviewer has is whether NB-1 and NB-SD therefore represent outliers amongst the neuroblastoma cases, whilst SK-N-AS and NGP might be more representative of typical levels of ALK expression. These concerns could be allayed in several ways: 1) the quantification of ALK expression in the targets used in the paper in terms of molecules per cell; 2) in vivo data from NGP-WT versus NGP-KO (was this done; if yes then including it would strengthen the paper even if the ADC has no significant benefit versus control); 3) inclusion of any additional animal studies in ALK positive targets where the ADC is ineffective; or a statement that there are not any; 4) mechanistic insight as to why SK-N-AS is resistant to the ADC despite showing ALK expression- it might be failure of internalisation in SK-N-AS related to avidity caused by low antigen expression compared to the other targets as shown in figure 4. Are there internalisation data on targets such as COG -N-424x or HCT-116 which looks to be similarly antigen dim but have high sensitivity and good in vivo efficacy.

Response: We have addressed quantification of ALK cell surface expression in our response to comment 4 above. We provide new *in vivo* experimental data for CDX0239-PBD, including tumor volume and survival curves for NGP ALK wild-type (WT, new panels in **Figure 6A and Figure 7A**) and NGP ALK knockout (KO, new panels in **Figure 6D and Figure 7D**) and have updated the Results section (**lines 284-293, lines 299-301, and lines 304-308**) and figure legends accordingly. These data demonstrate that although CDX0239-PBD and free PBD exhibits similar IC₅₀ values *in vitro* (42.97 pM and 31.08 pM, respectively), treatment of NGP ALK WT xenografts with CDX0239-PBD leads to complete and maintained responses with 100% survival, confirming that NGP has ALK surface expression above the therapeutic threshold for potent efficacy with CDX0239-PBD. In the NGP ALK KO study (n=5 mice/study arm), three xenografts treated with CDX0239-PBD demonstrated rapid tumor growth and two demonstrated an initial transient response with subsequent rapid tumor growth. To determine if the initial response to CDX0239-PBD in the two NGP ALK KO xenograft models was due to any residual ALK expression, western blot analysis for total ALK was completed on NGP ALK WT tumors treated with normal saline vehicle, NGP ALK KO tumors treated with normal saline vehicle, and NGP ALK KO tumors treated with CDX0239-PBD. We confirmed the presence of ALK expression in the NGP ALK WT tumors and confirmed the absence of ALK expression in the NGP ALK KO tumors (Figure included below).

We postulate that the initial transient response to CDX0239-PBD in two of the NGP ALK KO xenografts was due to non-specific internalization of CDX0239-PBD via binding of the Fc region on the full-length antibody to FCγRs (**Takai. Nat Rev Immunol. 2002. PMID: 12154377**). This effect has been previously demonstrated in full-length antibody-based ADCs similar to CDX0239-PBD (**Aoyama et al. Pharm Res. 2021. PMID: 34961908**). We note that this anti-tumor effect was transient as these NGP ALK KO tumors had eventual tumor growth velocity comparable to the control groups, suggesting that this effect was minimal and is supported by lack of weight loss or signs of toxicity due to non-specific binding and internalization of CDX0239-PBD in all the animal models. Future studies beyond the scope of this work could address this effect by including blocking antibodies to FCγRs with CDX0239-PBD treatment or leveraging antibody engineering approaches such as L234A/L235A substitutions to reduce Fc-mediated effector functions (**Aoyama et al. Pharm Res. 2021. PMID: 34961908**) and we have included this rationale in the Discussion section (**lines 413-418**). We further postulate in the discussion how we will investigate additional linker-payload chemistries including conjugation methods shielding the

linker-payload molecule within the Fc region to reduce hydrophobicity, aggregation, and non-specific internalization of the ADCs (**lines 418-420**).

Lastly, we postulate that the very low levels of ALK cell surface receptor density in SK-N-AS (shown via flow cytometry in **Figure 3**) are amenable to targeting with an ADC modality but are too low to induce complete and sustained tumor regressions. This is supported by the present but decreased internalization of CDX0239 by SK-N-AS cells (**Figure 4**) and the partial and transient anti-tumor activity observed *in vivo* (**Figure 6A**), which exhibits that CDX0239-PBD can selectively deliver PBD to cells with low levels of surface ALK expression with subsequent bystander cytotoxicity. Given COG-N-424x and HCT-116 demonstrate significantly higher ALK expression than SK-N-AS (but not nearly as high as NB-1), we conclude that these models are above the therapeutic threshold of ALK surface expression required for internalization of CDX0239-PBD that leads to objective and sustained anti-tumor activity. Additionally, we demonstrate superior internalization of CDX0239 in cell lines with moderate surface ALK expression (NB-SD and IMR-32, **Supplemental Figure 3**) as compared to SK-N-AS, further supporting that these tumor subtypes are above the therapeutic threshold of ALK surface expression necessary for adequate internalization of CDX0239-PBD and sustained anti-tumor activity. We elaborate on these data in the Discussion section (**lines 400-410**).

(b) Antibody controls with modifications to reduce efficacy are included in supplemental figures 3 and 4 but the text (results or figure legends) does not explain what the controls represent nor discuss relevance. Overall this reviewer is excited to see the data especially given the additional strong responses in the HCT, SW48 and RH41 models- but addressing the concerns above would strengthen the conclusion that the mechanism is ALK and internalisation dependent. Is there a simple threshold of expression for efficacy (which could imply a future predictive biomarker) or are there other factors at play related to ALK internalisation kinetics?

Response: We agree that we did not sufficiently describe the multiple antibody controls we designed with various modifications to support our conclusion that CDX0239-PBD demonstrates efficacy and specificity through ALK cell surface binding and internalization. Relevant antibody controls were previously listed in the Methods section (**lines 681-685**) which include human IgG, CDX0239, CDX0239 with deglycosylation via asparagine to alanine mutation, CDX0239 with deglycosylation via enzymatic digestion, and CDX0239 with deglycosylation via asparagine to alanine mutation with conjugated L6 linker.

- The human IgG control was added to demonstrate lack of anti-tumor activity of a non-target full-length human IgG.
- The CDX0239 control was utilized to demonstrate lack of anti-tumor activity of the unconjugated ALK-directed monoclonal antibody.
- CDX0239 with deglycosylation via asparagine to alanine mutation and CDX0239 with deglycosylation via enzymatic digestion were used to demonstrate lack of anti-tumor activity with maximization of antibody-dependent cellular cytotoxicity (ADCC) by enhancing antibody binding to FcγRIIIA (CD16) on leukocytes (**Trastoy et al. Nat Commun. 2023. PMID: 36973249**).
- Lastly, CDX0239 with deglycosylation via asparagine to alanine mutation with linker was included to demonstrate lack of linker-induced anti-tumor activity and confirm conjugation of the PBD dimer payload is required to induce cytotoxicity.

We now include relevance of antibody controls in the Results section (**lines 293-298**) and we have added additional interpretation of the results from our controls in the Discussion section (**lines 429-440**).

Comment 6. *Figure 8 shows the NB-SD model to be sensitive to ADC but resistant to lorlatinib. It would be important to know if the data in 8B and 8D is the same as 5A or an experimental repeat. They look very similar and so if a repeat, this would add to the story and should be made clear in the narrative.*

Response: The data in **Figure 8B** and **Figure 8D** are from the same *in vivo* study presented in **Figure 6A** and **Figure 7A**. The curves are redemonstrated in Figure 8 to highlight the efficacy of CDX0239-PBD in an ALK surface-expressing and ALK-

mutated model that demonstrates *de novo* resistance to small molecular inhibition of ALK, further emphasizing the application of CDX0239-PBD to patients with limited therapeutic options and emphasizing the proof-of-concept that the same antigen that is resistant to small molecular inhibition can be effectively targeted with an ADC regardless of mutational status. We have updated **Figure 8C** and **Figure 8D** with modification of the x-axis scale to weeks to remain consistent with survival curves throughout the manuscript.

REVIEWER 2:

The manuscript is well-structured, and the experiments are generally well designed and executed. However, there are a few aspects that require further clarification and additional data to strengthen the conclusions, particularly regarding the therapeutic potential of CDX0239-PBD in CNS malignancies.

Comment 1. *IC50 Determination: The authors should provide the IC50 value of CDX0239-PBD to allow for a better understanding of its potency.*

Response: IC₅₀ values were previously reported in the Results section (**lines 254-264**) and presented as a table for each cell line (**Supplemental Table 6**).

Comment 2. *Blood-Brain Barrier Considerations: Given that CDX0239-PBD would need to cross the blood-brain barrier to be effective against CNS malignancies, the expected concentration of the drug in the brain should be discussed. Experimental data or relevant pharmacokinetic modeling would strengthen this aspect.*

Response: Neuroblastoma is a disease of the peripheral nervous system, not the central nervous system (CNS). While neuroblastoma can metastasize to the brain, this is rare at diagnosis. With respect to primary CNS malignancies, there is precedent for ADCs having activity against CNS metastases. Trastuzumab deruxtecan is a HER2-directed full-length IgG-based ADC that has been studied in HER2-positive breast cancer with brain metastases and a single arm, Phase II clinical trial demonstrated complete and partial intracranial responses when administered intravenously at standard systemic dosing and is currently being studied in patient population with HER2 low expressing breast cancer with brain metastases (**Bartsch et al. Nature Medicine. 2022. PMID: 35941372**). Additionally, when thinking about CNS malignancies that could benefit from an ALK-directed ADC, we know that selective disruption of the blood-brain barrier (BBB) in some tumors with CNS metastases has been previously established (**Tiwary et al. Scientific Reports. 2018. PMID: 29844613**). Altogether, these studies suggest that CDX0239-PBD may have enhanced permeability into the CNS in the setting of brain metastases or primary CNS tumors as compared to a BBB unaffected by malignancy or systemic disease. We have now touched on this in the Discussion section (**lines 390-396**). Future studies beyond the scope of this work will evaluate CNS pharmacokinetics in relevant models.

Comment 3. *ALK Expression Assessment: The authors use ALK-D5F3 for immunohistochemical staining in FFPE TMA slides. The staining in Figure 1 appears to be strong, with a moderately high H-score. How does this compare to ALK-amplified or overexpressed tumors? Providing a comparative analysis would enhance the interpretation of these results.*

Response: To address this reviewer's questions and similar comments made by other reviewers, we have now comprehensively annotated our neuroblastoma tumor tissue microarray (TMA) which includes 55 tumors representative of patients with this disease: seven tumors have an underlying *ALK* mutation, and all remaining tumors have wild-type *ALK* with a range of protein expression. The clinical, histopathologic, and molecular features of the tumors embedded in this TMA are representative of the neuroblastoma patient population including diversity in MYCN amplification status, *ALK* status, sex, age group, differentiation, histology, ploidy status, mitosis-karyorrhexis index, timing of biopsy (diagnostic, post-chemotherapy, and at relapse), Neuroblastoma Risk Group Staging System (INRGSS) risk stratification, and International Neuroblastoma Risk Group (INRG) stage. None of these features demonstrate statistically significant differences in average H-scores for ALK staining, supporting the conclusion that most neuroblastoma tumors express ALK. We have updated Figure 2 and our results section to include a comprehensive annotation of the TMA H-score analysis by subgroups (**lines 175-188 and Figure 2B**). Additionally, we have updated Figure 2 to include a high-powered pictograph (20x magnification) of a diagnostic neuroblastoma tumor core with an H-score of 142.33 which represents the near average H-score of 144.71 across the TMA (**lines 188-189, Figure 2C**).

The reviewer specifically asks how TMA staining for ALK compares to expression in *ALK*-amplified tumors. Notably, just as *ALK* amplification is rare in patients (~3% of patients with high-risk neuroblastoma, **Berko et al. J Clin Oncol. 2025. PMID: 40036726**), none of the tumors on this TMA harbor *ALK*-amplification. The tumors represent a mixture of *ALK* wild-type and *ALK*-mutant tumors and demonstrate the spectrum of ALK protein expression. *ALK* amplification in human neuroblastoma tumors denotes an increased in the number of copies of the *ALK* gene (gene dosage), typically ranging from 20-100+, which consequently results in dramatic overexpression at the protein level as demonstrated with the NB-1 cell line, one of the only existing models of *ALK* amplification. We previously quantified cell surface ALK expression in NB-1 and a panel of other human neuroblastoma-derived cell lines using quantitative flow cytometry with a chimeric anti-ALK antibody (**Sano et al. Sci Transl Med. 2019. PMID: 30867324**). As expected, NB-1 demonstrated significantly higher ALK cell surface expression when compared with all other ALK wild-type cell lines. This suggests that *ALK*-amplified tumors would likely have H-scores closer to the maximum of 300, and that the average H-score of 144.71 in our patient tumor TMA is representative of ALK overexpression across the majority of neuroblastoma tumors. We clarify the nuances of ALK expression (wild-type versus amplification) in the Discussion section (**line 402-408**).

Comment 4. *Expression Threshold for Efficacy: At what expression level (e.g., copy number threshold) does CDX0239-PBD lose efficacy? This information is critical for understanding patient selection criteria.*

Response: We agree with the reviewer that determining the ALK expression threshold for efficacy is of great importance to evaluate the translational potential of CDX0239-PBD and to serve as a potential biomarker for patient selection. Important to note that “expression level” is directly linked to receptor density on the cell surface. A very small subset of tumors, as noted above, will have true copy number amplification. We explain in detail in response to *Reviewer 1, Comment 4*, why we were unable to conjugate a fluorophore to CDX0239 for quantitative flow cytometric analysis of receptor density (antibodies bound per cell).

Our existing flow cytometry data enabled us to calculate the ALK:IgG geometric mean of ALK surface expression in comparison to non-specific binding of the IgG control in each xenograft model resulting in an effective quantitative representation of ALK surface expression for each xenograft model. We show that CDX0239-PBD leads to complete maintained responses in the NGP *ALK* wild-type (WT) xenograft model (ALK:IgG geometric mean of 8.6), and as expected did not demonstrate activity in the NGP *ALK* knockout (KO) model (ALK:IgG geometric mean of 1.6). CDX0239-PBD has decreased efficacy in the SK-N-AS xenograft (ALK:IgG geometric mean of 3.5). Thus, we can conclude that the therapeutic threshold of ALK surface expression for CDX0239-PBD efficacy lies between an ALK:IgG geometric mean of 3.6-8.6. While immunohistochemistry (IHC) is a more practical method for assessing ALK expression in the clinical setting (as we will not have access to fresh tumors for flow cytometry), it is fraught with many challenges. The extensive data we have for ALK staining in both a neuroblastoma patient-derived xenograft (PDX) tissue microarray (TMA, **Sano et al. Sci Transl Med. 2019. PMID: 30867324**) and a human neuroblastoma tumor TMA in this body of work suggests that most patients with neuroblastoma could benefit from an ALK-directed ADC and that the clinical trial will teach us more precisely about the expression threshold and response. We elaborate on expression threshold and efficacy in the Discussion section (**lines 398-410**).

Comment 5. *Drug-Antibody Ratio (DAR) and Pharmacokinetics: The manuscript states a DAR of 6 for CDX0239-PBD. The authors should provide the in vivo pharmacokinetic (PK) profile, including measurements of free PBD levels, to evaluate potential systemic toxicity and drug clearance.*

Response: PBD is a highly potent DNA crosslinking agent that would induce systemic and end-organ toxicity when in circulation. While we did not measure free PBD levels, we know that CDX0239 has cross species reactivity with human and murine ALK (**Supplemental Figure 1**), allowing us to surveil for on and off-target toxicity. We did not observe significant decreases in mouse weights or signs of toxicity during or after treatment with CDX0239-PBD in any of the xenograft models throughout the study (**lines 308-309 and lines 326-327, Supplemental Figure 6**). These data support stability of the linker and lack of systemic free PBD. When thinking about pharmacokinetic (PK) studies, the two analytes of interest for ADCs are total antibody and conjugate. The development of these analytes is beyond the scope of this work, particularly since the pivotal PK studies needed for Investigational New Drug (IND)-enabling studies will have to be performed and measured in cynomolgus monkeys for prediction of human PK and prior to the initiation of a clinical trial. We have further elaborated about the importance of the DAR, PK, and measurements of free drug in the Discussion section (**lines 485-488**).

Comment 6. *In Vivo Study Design: The efficacy argument for CDX0239-PBD could be further strengthened by including an additional treatment arm with PBD alone at equimolar concentrations. This would help distinguish the contribution of the ADC from that of the free payload.*

Response: PBD is a potent DNA crosslinking agent with significantly higher potency than other DNA-binding agents that are administered systemically such as cyclophosphamide and cisplatin, and PBD has been precluded from systemic administration due to the significant cardiotoxicity profile in animal models due to hydroxylation at C-9 (Li et al. **Appl Environ Microbiol.** 2009. PMID: 19270142). Therefore, free PBD cannot be given safely, nor at relevant doses, via systemic administration in murine models as it would include significant cardiotoxicity in addition to end-organ toxicity and myelosuppression. Notably, CDX0239 has cross-species reactivity with human and murine ALK (**Supplemental Figure 1**). To distinguish the contribution of the ADC from that of any off-target activity, we **i**) demonstrate lack of cytotoxicity in co-culture with the NGP ALK knockout (KO) cell lines (**Figure 5C**), **ii**) performed an *in vitro* competition study in the ALK-amplified NB-1 cell line demonstrating rescue of neuroblastoma cell viability when unconjugated CDX0239 is added to co-culture with CDX0239-PBD for 5 days (**Figure 5E**), and **iii**) demonstrate complete and sustained tumor regressions and 100% survival in our ALK-expressing xenograft models treated with CDX0239-PBD without significant weight loss or signs of toxicity (**Figures 6-8, Supplemental Figure 6**). These data support linker stability without off-target cleavage and subsequent off-target toxicity.

Comment 7. *Typographical Error: Line 332 contains a minor typo where “am” should be corrected to “an.”*

Response: Corrected.

REVIEWER 3:

*In this manuscript, A. Guerra et al. describe a preclinical evaluation of CDX0239-PBD, a novel antibody-drug conjugate (ADC) targeting Anaplastic Lymphoma Kinase (ALK), a lineage-restricted antigen overexpressed in neuroblastoma (NB) and several other pediatric and adult malignancies. Utilizing a humanized ALK-specific monoclonal antibody conjugated to a pyrrolobenzodiazepine (PBD) payload, the study demonstrates potent and selective cytotoxicity in ALK-expressing tumor cells, with minimal off-target effects in normal tissues. The authors provide extensive *in vitro* and *in vivo* evidence of ALK surface expression-dependent internalization, lysosomal trafficking, and DNA-damage-mediated apoptosis, leading to sustained tumor regression in multiple xenograft models. These findings suggest that CDX0239-PBD is a promising candidate as a precision therapeutic for ALK-expressing cancers. Overall, the study represents a significant advance in targeted therapies for NB and other ALK-positive malignancies. The experimental design is rigorous, with appropriate positive and negative controls, *in vitro* and *in vivo* validation, and the use of multiple ALK-expressing cancer models. Notably, demonstrated minimal ALK expression in normal tissues decreases the risk of CDX0239-PBD on-target toxicity. Mechanistically, the results demonstrate that the ADC works via ALK internalization and lysosomal localization leading to induction of apoptosis in cancer cells. However, there are the following concerns:*

Comment 1. *The reliance on transplantable xenograft models limits the understanding of CDX0239-PBD efficacy against spontaneous tumors with their complex TME. Authors could try a syngeneic NB model or at least acknowledge the limitations of xenograft models in discussion.*

Response: We acknowledge the limitations of understanding the efficacy of CDX0239-PBD using subcutaneously implanted xenograft models as compared to spontaneous tumors such as a TH-MYCN transgenic murine model. As suggested, we have added discussion on future utilization of spontaneous tumor models to better recapitulate the complex TME to the Discussion section (**lines 480-483**) and plan these studies as part of our IND enablement of the clinical candidate we will create.

Comment 2. *Figure 2 lacks an analysis of ALK surface expression's association with MYCN amplification and patient age, independent prognostic markers in NB.*

Response: We have updated **Figure 2** and the results section to now include a detailed annotation and analysis of ALK staining H-scores in our neuroblastoma tumor tissue microarray (TMA) by MYCN amplification status, ALK status (wild-type and mutated), sex, age group, differentiation, histology, ploidy status, mitosis-karyorrhexis index (MKI), time of biopsy (diagnostic, post-chemotherapy, and at relapse), revised International Neuroblastoma Risk Group Staging System (INRGSS) risk stratification, and International Neuroblastoma Risk Group (INRG) stage in comparison to ganglioneuroblastoma tumors and healthy tissues (placenta, adrenal cortex, adrenal medulla, and tonsil, **lines 175-188, Figure 2B**). Our analysis demonstrates significant ALK protein expression across all subsets (average H-score of 144.71 among 55 neuroblastoma tumors) with no significant differences in ALK staining H-score when compared between genomic, tumor biology, demographic, and diagnostic subgroups.

Comment 3. *Although the study discusses ADCs' ability to overcome TME-related barriers, more in-depth exploration of how CDX0239-PBD penetrates tumor vs. normal tissues (e.g. liver) would strengthen the translational potential.*

Response: We provide a more in-depth explanation of how CDX0239-PBD might overcome TME-related barriers including the extracellular matrix (ECM) and the highly immunosuppressive environment in neuroblastoma in the Discussion section (**lines 380-390**). ADCs can overcome these barriers by enabling bystander cytotoxicity to tumor cells with low surface target antigen (Li et al. **Cancer Res. 2016. PMID: 26921341**) in addition to tumor cell death with subsequent release of damage-associated molecular patterns (DAMPs) to stimulate dendritic cell activation that facilitate antigen uptake and migration to the lymph nodes, ADC-mediated reduction of Tregs, augmentation of MHC class I expression to promote activity of cytotoxic T cells, and enhancement of leukocyte infiltration, expansion, and interferon gamma (IFN γ) production (Chang et al. **J Clin Invest. 2023. PMID: 37712425**). Our *in vivo* studies demonstrate potent efficacy of CDX0239-PBD with complete maintained responses in moderate to high surface ALK-expressing xenograft models and demonstrate an initial transient response in low surface ALK-expressing xenograft models suggesting the presence of bystander cytotoxicity that overcomes structural and spatial barriers of the TME in addition to cytotoxicity in tumor cells with low surface target antigen. We will explore additional TME-directed and immune-enhancing properties of our ALK-directed ADC further in future studies with spontaneous tumor and immunocompetent animal models. We address this in depth in the Discussion section (**lines 402-410 and lines 480-483**).

REVIEWER 4:

The submitted manuscript, "A humanized ALK-directed antibody-drug conjugate CDX0239-PDB demonstrates potent efficacy in ALK-expressing cancers" by Guerra and colleagues, presents a compelling approach to precision chemotherapy development. Over more than a decade, the authors have advanced this research toward a potential clinical application. Beginning with a murine antibody capable of blocking ALK activity, the researchers have systematically improved its properties through modifications, enhancing its affinity and specificity. Ultimately, they generated a humanized ALK-directed monoclonal antibody with high affinity, conjugated to pyrrolobenzodiazepine. The manuscript presents promising results, demonstrating significant responses in ALK-expressing cell line xenografts and opening a viable path for clinical development.

Comment 1. *Lines 95–96: It seems odd that reference 2 appears both in the middle and at the end of the sentence. The authors are making a point, but scientists in the field are already aware of these facts.*

Response: We have now removed the citation of reference 2 from the middle of the sentence and retained its citation at the end of the sentence with the additional cited studies which together demonstrate ALK as the major cause of hereditary neuroblastoma and as the gene with the most common somatic single nucleotide mutations in the sporadic forms of the disease.

Comment 2. *Lines 104–106: The reference is incorrect. Where has this manuscript been published? The authors should also include additional references, as this observation has been reported elsewhere, such as in PMID: 20719933, PMID: 26121087, and PMID: 25071110.*

Response: The study referenced has now been published in the Journal of Clinical Oncology (**Berko et al. J Clin Oncol. 2025. PMID: 40036726**) and the reference has been updated accordingly. We have added additional references in addition to our own recent publication and highlight that MYCN amplification is more frequent with ALK aberrant tumors, particularly those harboring ALK F1174 mutations. Previous studies have demonstrated higher frequencies in ALK and RAS-MAPK pathway mutations at relapse, suggesting an enrichment of selected subclonal mutations as a mechanism of therapy resistance (**lines 109-112**).

Comment 3. *Line 110: The authors should add another reference to emphasize that most patients harboring neuroblastoma tumors express ALK (PMID: 38229432).*

Response: We have added this reference to this statement to further emphasize that native ALK protein is expressed on the cell surface of most neuroblastoma tumors (**line 116**).

Comment 4. *Lines 120–125: This sentence could be improved for clarity, making it more accessible to general readers. Additionally, several long sentences throughout the manuscript should be refined for readability.*

Response: We have modified this statement and multiple statements throughout the manuscript to improve clarity.

Comment 5. *Line 161: Add reference PMID: 38229432, which documents ALK expression in neuroblastoma patients. This should be included, as the authors validate previously published findings effectively.*

Response: This reference has now been added to this statement (**line 173**)

Comment 6. *Line 184: While the methods section clarifies this to some extent, the authors should provide more details on the methodology and procedure for generating CDX0239. Was CDX0239 derived from the discovery of CDX125, as referenced in reference 31, or from PMID: 30867324? Additionally, where was the murine anti-ALK antibody originally obtained?*

Response: We have expanded the methods section to provide additional information on the origin, methodology, and procedure for generating humanized CDX0239 which was derived from the discovery of the chimeric antibody CDX0125 (**lines 549-568**).

Comment 7. *Line 185: The authors use the bio-layer interferometry method. It would be appropriate to cite a more foundational article describing this technique, such as PMID: 22414486.*

Response: This foundational study is now included in our citations.

Comment 8. *Lines 185–203: Is CDX0239 activating or inhibiting ALK activity?*

Response: Previous studies have demonstrated that the chimeric ALK-directed monoclonal antibody CDX123 is an ‘activating’ antibody that likely functions by crosslinking of receptor monomers without affecting ligand recognition, and the chimeric antibody CDX0125 is an ‘inhibitory’ antibody leading to decreased ALK phosphorylation due to binding overlap at the ALKAL binding site (**Li et al. Nature. 2022. PMID: 34819665; Sano et al. Sci Transl Med. 2019. PMID: 30867324**). The ‘activating’ properties of CDX123 and the ‘inhibitory’ effects of CDX0125 were demonstrated in the

neuroblastoma cell line IMR-32 (*ALK* wild-type) which does not harbor an *ALK* mutations and requires ligand binding for activation. CDX0239 is a humanized antibody derived from the discovery of the chimeric antibody CDX0125. We demonstrate that CDX0239 does not have an effect on phosphorylated *ALK* levels in the *ALK*-mutated (c.3522C>A, F1174L) NB-SD xenograft (**Harenza et al. *Sci Data*. 2017. PMID: 28350380**). Given NB-SD is *ALK*-mutated, there is constitutive *ALK* activation without the presence of a ligand compared to wild-type *ALK*, we would not expect CDX0239 to induce decreased levels of phosphorylated *ALK* even if CDX0239 is bound to the *ALK* binding site. This is suggested by no evidence of decreased phosphorylated *ALK* expression after treatment with CDX0239-PBD (**Figure 8E**). Additionally, we demonstrate no evidence of anti-tumor activity with unconjugated CDX0239 alone in the *ALK*-amplified NB-1 neuroblastoma cell line and matched xenograft model (**Figure 5B**). If CDX0239 had a significant inhibitory effect on phosphorylated *ALK* signaling, there would likely be an observed anti-tumor activity as previously demonstrated with small molecular inhibition of *ALK* in NB-1 (**Infarinato et al. *Cancer Discovery*. 2016. PMID: 26554404**). This is now clarified in the Discussion section (**lines 429-436**). Lastly, we demonstrate that binding of CDX0239 to surface *ALK* leads to endocytosis of the *ALK* receptor-antibody complex with intracellular localization to the lysosome (**Figure 4 and Supplemental Figure 3**) which leads to a reduction in total *ALK* expression (**Figure 8E**).

Comment 9. Line 313: CDX0239-PBD did not affect phosphorylated *ALK* (p*ALK*1604) levels. What about phosphorylated p*ALK*1278, which is necessary for *ALK* activity? The authors should verify their data using a p*ALK*1278 antibody. Have they observed any additional differences in downstream *ALK* signaling when using CDX0239? This is not addressed in the manuscript.

Response: We and others have shown that there is variability in expression of the two key *ALK* phospho-tyrosine sites, namely p*ALK*128 and p*ALK*1604, and this differs by model and underlying activating *ALK* mutation. We attempted numerous times to analyze the processed NB-SD tumor lysates for p*ALK*1278 expression, however we were unable to detect this phospho-site by Western Blot. NB-SD harbors a c.3522C>A, F1174L mutation which confers constitutive activation with aberrant downstream signaling including the PI3K/AKT, RAS/MAPK, and JAK/STAT pathways (**Harenza et al. *Sci Data*. 2017. PMID: 28350380; Debruyne et al. *Oncogene*. 2015. PMID: 26616860**). To demonstrate any potential effect of CDX0239 on downstream *ALK* signaling, we analyzed the NB-SD tumor lysate samples for phosphorylated BRAF (p445, Cell Signaling, #2696S) as BRAF is downstream of *ALK* in the RAS/MAPK signaling pathway. We found decreased expression of pBRAF 7 days after treatment with CDX0239-PBD (Figure included below) which is consistent with our previous results demonstrating diminished total *ALK* expression 7 days after treatment (**Figure 8E**). Given that there was no effect on p*ALK*1604 expression after treatment with CDX0239-PBD, we postulate that the diminished expression of phosphorylated BRAF 7 days after treatment with CDX0239-PBD is due to antibody-receptor internalization which leads to diminished total *ALK* expression and thus diminished signaling of downstream proteins. Further characterization of additional signaling proteins downstream of *ALK* is out of scope for this study, particularly since *ALK* signaling within the context of activating mutations in the full-length receptor remains undefined.

Comment 10. Line 398: The authors should include reference PMID: 34646012, as it was published in the same issue as reference 59.

Response: We have added this reference.

Comment 11. Line 453: The specificity and selection process for the humanized antibody (CDX0239) are unclear. Which

ALK epitope does it bind to? Is it ALK or ALK/ALKAL2? The authors refer to reference 31, where they express a codon-optimized cDNA encoding the extracellular glycine-rich domain of ALK (aa 678-1030) fused to the ALK ligand ALKAL2 (aa 71-152). Could it be that the antibody only recognizes ALK RTK when bound to an ALK ligand? The authors should clarify this.

Response: We have added more details to describe the selection process for the humanized antibody CDX0239. The original ALK-directed murine antibody was generated against human ALK (T637-S1038) comprising the glycine rich and epidermal growth factor domain as stated above (**Sano et al. *Sci Transl Med.* 2019. PMID: 30867324**). Although the crystal structure of the ALK extracellular domain bound to CDX0239 is not available, the chimeric version of the anti-ALK antibody (CDX0125) was shown to protect a peptide in the C-terminal loop of the TNF- α like region of ALK (Y966-V972) which overlaps with the activating peptide (ALKAL) binding site in hydrogen/deuterium exchange mass spectrometry (HDX-MS; **Li et al. *Nature.* 2021. PMID: 34819665; Figure 4**), indicating that the antibody binds to the extracellular domain proximal to the ALK cleavage site (N654-L655; **Huang et al. *Cell Rep.* 2021. PMID: 34260934**). Our studies demonstrate that CDX0239 binds to the receptor tyrosine kinase (RTK) in the absence of an ALK ligand as we successfully utilized CDX0239 to bind surface ALK to determine surface ALK expression via flow cytometry (**Figure 3**). We have further clarified this in the manuscript in the Methods section (**lines 549-561**).

REVIEWER 1

Comment 1. *Figure 2; Immunohistochemistry uses antibody clone D5F3 but neither text nor methods informs the reader which epitope this antibody binds and if the sections are being scored for total expression or membrane staining. In a quick search of literature I could not find information on the position of the epitope. This is of particular relevance given the known cleavage of ALK in neuroblastoma. If the D5F3 epitope were N terminal to the cleavage site then the staining intensity of neuroblastoma is likely to be under-represented.*

Response: We have now provided additional information about antibody clone D5F3 in the manuscript (**lines 172-173 and lines 525-527**). Rabbit anti-ALK D5F3 (Cell Signaling, #3633) is an FDA and Clinical Laboratory Improvement Amendments (CLIA)-approved antibody as a standalone test for ALK rearrangements in lung cancer. While the binding epitope is not available in the literature as it is proprietary, it is known that D5F3 binds to the cytoplasmic kinase domain of ALK and hence is able to recognize ALK fusion proteins which are lacking the extracellular, N-terminus domain of ALK (**Sung and Kim. Thoracic Cancer. 2024. PMID: 39257160**). As such, this antibody provides interpretable and broadly relevant staining intensity of total full length ALK in our neuroblastoma tissue microarray (TMA) samples.

Comment 2. *Figure 2C-D; staining in pituitary is described as equivocal but it is hard for the reader to appreciate what this means. Does this mean some rare cells with bright membrane staining or all cells with faint and or indeterminate staining which could be for example cross reactivity with a cytoplasmic antigen. There is not enough detail in results or methods section to determine this and the resolution of figure 2D, at least in the pdf provided to reviewers, is too low to appreciate the staining.*

Response: We now define equivocal staining as “minimally present staining of a cell with anti-ALK D5F3 at 1:500 dilution” and we previously defined the calculation the total H-score for each tumor or tissue sample based on the percent of cells with the most prominent intensity (1 representing equivocal, 2 weak, and 3 strong positive staining; **lines 537-540**). We have updated Figure 2 to include a high-powered pictograph (20x magnification) of a diagnostic neuroblastoma tumor core from our patient TMA stained with anti-ALK D5F3 (**Figure 2C**) for qualitative comparison to our pituitary samples (**Figure 2E**) that demonstrated equivocal staining in a subset of cells, and this is now further clarified in the Results section (**lines 193-197**). The represented diagnostic neuroblastoma core example in **Figure 2C** is from a right neck mass diagnostic biopsy from a 9-year-old male with high-risk neuroblastoma (*MYCN* non-amplified and *ALK* wild-type). This core was selected for representation as it has an H-score of 142.33 which reflects the near average H-score of 144.71 for ALK staining in the neuroblastoma tumor TMA (**Figure 2C**).

Comment 3. *Figure 3. There is inadequate information to assess the cell staining. The samples are described as xenografts but there is no section in the methods on in vivo tumour growth; is it subcutaneous or orthotopic? What size of tumour? – these factors can influence the amount of stroma that needs to be gated out. What gating was used to gate out myeloid and endothelial cells, doublets etc? Without the gating information can we exclude ALK staining on stroma? Most importantly there is no information on CDX0239 epitope. Is it membrane proximal or distal to the ALK cleavage site? That is of utmost importance since if CDX0239 is membrane distal to the cleavage then cells with cleaved ALK may become treatment escape variants.*

Response: We agree that variables such as tumor size and the site of xenograft implantation can influence ALK surface expression and overall efficacy of ALK-directed therapies. We clarify that tumors were implanted subcutaneously into the right flank of female CB17 SCID mice and randomized into treatment groups when tumors reached 0.20-0.25 cm³ and treated via intraperitoneal injection weekly for three consecutive weeks at 1 mg/kg of defined treatment or 10 mL/kg of normal saline vehicle (**lines 679-685**). Untreated parental tumors were assessed for ALK surface expression after reaching volumes of 0.5-1.0 cm³, and we have now included details on xenograft tumor size before flow cytometry analysis for surface ALK expression (**line 613**). We employed SCID mice for our *in vivo* studies and utilized a mouse cell depletion kit (Miltenyi Biotec, #130-104-694) to remove mouse myeloid, endothelial, and stromal cells to eliminate any confounding background staining. Additionally, we have included an example of our gating strategy used for analysis of our flow cytometry data after mouse cell depletion (**Supplemental Figure 2**).

Regarding the CDX0239 epitope, the original ALK-directed mouse monoclonal antibody was generated against human ALK comprising the glycine rich and epidermal growth factor domain (T637-S1038; **Sano et al. Sci Transl Med. 2019. PMID: 30867324**). Although the crystal structure of the ALK extracellular domain bound to the antibody is not available, the chimeric version of the ALK-directed antibody (CDX0125) was shown to protect a peptide in the C-terminal loop of the TNF- α like region of ALK (Y966-V972) which overlaps with activating peptide (ALKAL) binding site in hydrogen/deuterium exchange mass spectrometry (HDX-MS; **Li et al. Nature. 2021. PMID: 34819665, Figure 4**), indicating that the antibody binds to the extracellular domain proximal to the ALK cleavage site (N654-L655; **Huang et al. Cell Rep. 2021. PMID: 34260934**). The antibody used for ADC synthesis (CDX0239) was humanized via homology remodeling of the DNA sequence for the variable regions to match human IgG1, yielding 12 heavy and 8 light chains, and 96 combinations. Plasmids encoding the humanized anti-ALK IgG1 antibody were cloned in Expi239 cells to produce the humanized anti-ALK monoclonal antibody and is now described in detail in the manuscript (**lines 549-564**). Importantly, for the determination of off-target binding and toxicity of CDX0239-PBD, we previously confirmed species cross-reactivity of CDX0239 with murine spontaneously occurring neuroblastoma tumors (**lines 203-207 and Supplemental Figure 1**).

Comment 4. *The data in figure 3 is also of great importance to evaluate translational significance of subsequent in vivo studies. The NB-1 tumour model shows what looks like 3 log higher MFI than the isotype control. This appears to be very significantly higher than typical flow plots of neuroblastoma using other antibodies (for example Walker et al PMID 28676342). Of course this might be because of relative brightness of the respective fluorophores used. Hence, it would be of great value to provide objective antigen expression data for the targets used in the paper using for example Quantibright beads.*

Response: We agree with the reviewer that determination of ALK cell surface expression is of great importance to evaluate the translational potential of CDX0239-PBD. The NB-1 xenograft model used in our study harbors high-level *ALK*-amplification, with >60 copies of the *ALK* gene as previously demonstrated, which at the protein level is robustly overexpressed (**Bresler et al. Sci Transl Med. 2012. PMID: 22072639**). This model represents the 3-4% of patients with high-risk neuroblastoma whose tumors harbor *ALK*-amplification (**Berko et al. J Clin Oncol. 2025. PMID: 40036726**). The COG-N-424x, NB-SD, and NGP neuroblastoma xenograft models demonstrate moderate ALK cell surface expression (**Figure 3A**), which is much more representative of most neuroblastoma tumors. These xenograft models demonstrate geometric means for surface ALK expression on flow cytometry that are similar to other xenograft models of neuroblastoma (**Walker et al. Mol Ther. 2017. PMID: 28676342, Figure 1B**). Importantly, we utilized the same humanized ALK-directed monoclonal antibody (CDX0239) for flow cytometry analysis of ALK surface expression and for synthesis of CDX0239-PBD and demonstrate that our estimated ALK surface expression by ALK:IgG geometric mean is positively correlated with efficacy of CDX0239-PBD. This is now clarified in the Discussion section (**lines 400-408**).

We attempted to conjugate a fluorophore to CDX0239 for Quantibrite™ bead (BD Biosciences, #340495) analysis. Quantibrite™ beads are intended for estimation of antibodies bound per cell (ABC) with fluorophore-conjugated antibodies as the beads link directly to the conjugated fluorophore. We were unable to successfully conjugate a fluorophore to CDX0239 despite multiple attempts as conjugation of a fluorophore led to loss of CDX0239 affinity for surface ALK. The effect of decreased antibody affinity after conjugation of fluorophores has been previously demonstrated (**Szabó et al. Biophys J. 2018. PMID: 29414714**). A quantification protocol utilizing a QIFKIT assay (Agilent Dako, #K007811-8) would be an alternative approach for quantification, however this kit is only available for mouse monoclonal antibodies which would not be compatible with our humanized ALK-directed antibody CDX0239. Additionally, full-length IgG humanized antibodies such as CDX0239 have non-specific binding to human cells despite utilizing an Fc block and bovine serum albumin (BSA) block. The non-specific binding of our human IgG control varied substantially between xenograft models but was consistent within xenograft models and enabled us to calculate the ALK:IgG geometric mean of ALK surface expression in comparison to non-specific binding of the IgG control in each xenograft model. This resulted in an effective quantitative representation of ALK surface expression for each xenograft model. We now detail this in the Results section along with previously stated ALK:IgG geometric means (**lines 207-224 and Supplemental Table 5**).

Comment 5. *Figures 5-7. The efficacy of the CDX0239-PBD ADC is shown in figures 5-7 with in vitro in 5 and in vivo in 6+7. In figure 5 the important control of NGP targets versus NGP knock-out is shown; this convincingly shows that in vitro cytotoxicity is ablated once ALK surface expression by genetic ablation, which would be strongly suggestive that the ADC activity is dependent on specific antibody to antigen interaction. In 5C, the PBD sensitivity is shown in molar values and appears to show equivalent sensitivity of NGP cells towards free PBD versus CDX0239-PBD. However, the other target lines*

NB-1 and NB-SD show higher sensitivity to the ADC than to free PBD and this maybe correlates with the somewhat higher expression of ALK in NB-1 and NB-SD than in NGP. The data suggests that targets such as SK-N-AS have a subthreshold ALK expression resulting in resistance to the ADC whilst NB1 and NB-SD are suprathreshold and are sensitive. NGP is more complex because of the equivalent IC50 to free payload and ADC; this maybe is related to intermediate ALK expression.

Response: We agree that it is of immense importance to address the ALK cell surface expression-dependent therapeutic threshold in relation to anti-tumor activity of CDX0239-PBD. We have addressed these concerns in detail below:

(a) The concern this reviewer has is whether NB-1 and NB-SD therefore represent outliers amongst the neuroblastoma cases, whilst SK-N-AS and NGP might be more representative of typical levels of ALK expression. These concerns could be allayed in several ways: 1) the quantification of ALK expression in the targets used in the paper in terms of molecules per cell; 2) in vivo data from NGP-WT versus NGP-KO (was this done; if yes then including it would strengthen the paper even if the ADC has no significant benefit versus control); 3) inclusion of any additional animal studies in ALK positive targets where the ADC is ineffective; or a statement that there are not any; 4) mechanistic insight as to why SK-N-AS is resistant to the ADC despite showing ALK expression- it might be failure of internalisation in SK-N-AS related to avidity caused by low antigen expression compared to the other targets as shown in figure 4. Are there internalisation data on targets such as COG -N-424x or HCT-116 which looks to be similarly antigen dim but have high sensitivity and good in vivo efficacy.

Response: We have addressed quantification of ALK cell surface expression in our response to comment 4 above. We provide new *in vivo* experimental data for CDX0239-PBD, including tumor volume and survival curves for NGP ALK wild-type (WT, new panels in **Figure 6A and Figure 7A**) and NGP ALK knockout (KO, new panels in **Figure 6D and Figure 7D**) and have updated the Results section (**lines 284-293, lines 299-301, and lines 304-308**) and figure legends accordingly. These data demonstrate that although CDX0239-PBD and free PBD exhibits similar IC₅₀ values *in vitro* (42.97 pM and 31.08 pM, respectively), treatment of NGP ALK WT xenografts with CDX0239-PBD leads to complete and maintained responses with 100% survival, confirming that NGP has ALK surface expression above the therapeutic threshold for potent efficacy with CDX0239-PBD. In the NGP ALK KO study (n=5 mice/study arm), three xenografts treated with CDX0239-PBD demonstrated rapid tumor growth and two demonstrated an initial transient response with subsequent rapid tumor growth. To determine if the initial response to CDX0239-PBD in the two NGP ALK KO xenograft models was due to any residual ALK expression, western blot analysis for total ALK was completed on NGP ALK WT tumors treated with normal saline vehicle, NGP ALK KO tumors treated with normal saline vehicle, and NGP ALK KO tumors treated with CDX0239-PBD. We confirmed the presence of ALK expression in the NGP ALK WT tumors and confirmed the absence of ALK expression in the NGP ALK KO tumors (Figure included below).

We postulate that the initial transient response to CDX0239-PBD in two of the NGP ALK KO xenografts was due to non-specific internalization of CDX0239-PBD via binding of the Fc region on the full-length antibody to FCγRs (**Takai. Nat Rev Immunol. 2002. PMID: 12154377**). This effect has been previously demonstrated in full-length antibody-based ADCs similar to CDX0239-PBD (**Aoyama et al. Pharm Res. 2021. PMID: 34961908**). We note that this anti-tumor effect was transient as these NGP ALK KO tumors had eventual tumor growth velocity comparable to the control groups, suggesting that this effect was minimal and is supported by lack of weight loss or signs of toxicity due to non-specific binding and internalization of CDX0239-PBD in all the animal models. Future studies beyond the scope of this work could address this effect by including blocking antibodies to FCγRs with CDX0239-PBD treatment or leveraging antibody engineering approaches such as L234A/L235A substitutions to reduce Fc-mediated effector functions (**Aoyama et al. Pharm Res. 2021. PMID: 34961908**) and we have included this rationale in the Discussion section (**lines 413-418**). We further postulate in the discussion how we will investigate additional linker-payload chemistries including conjugation methods shielding the

linker-payload molecule within the Fc region to reduce hydrophobicity, aggregation, and non-specific internalization of the ADCs (**lines 418-420**).

Lastly, we postulate that the very low levels of ALK cell surface receptor density in SK-N-AS (shown via flow cytometry in **Figure 3**) are amenable to targeting with an ADC modality but are too low to induce complete and sustained tumor regressions. This is supported by the present but decreased internalization of CDX0239 by SK-N-AS cells (**Figure 4**) and the partial and transient anti-tumor activity observed *in vivo* (**Figure 6A**), which exhibits that CDX0239-PBD can selectively deliver PBD to cells with low levels of surface ALK expression with subsequent bystander cytotoxicity. Given COG-N-424x and HCT-116 demonstrate significantly higher ALK expression than SK-N-AS (but not nearly as high as NB-1), we conclude that these models are above the therapeutic threshold of ALK surface expression required for internalization of CDX0239-PBD that leads to objective and sustained anti-tumor activity. Additionally, we demonstrate superior internalization of CDX0239 in cell lines with moderate surface ALK expression (NB-SD and IMR-32, **Supplemental Figure 3**) as compared to SK-N-AS, further supporting that these tumor subtypes are above the therapeutic threshold of ALK surface expression necessary for adequate internalization of CDX0239-PBD and sustained anti-tumor activity. We elaborate on these data in the Discussion section (**lines 400-410**).

(b) Antibody controls with modifications to reduce efficacy are included in supplemental figures 3 and 4 but the text (results or figure legends) does not explain what the controls represent nor discuss relevance. Overall this reviewer is excited to see the data especially given the additional strong responses in the HCT, SW48 and RH41 models- but addressing the concerns above would strengthen the conclusion that the mechanism is ALK and internalisation dependent. Is there a simple threshold of expression for efficacy (which could imply a future predictive biomarker) or are there other factors at play related to ALK internalisation kinetics?

Response: We agree that we did not sufficiently describe the multiple antibody controls we designed with various modifications to support our conclusion that CDX0239-PBD demonstrates efficacy and specificity through ALK cell surface binding and internalization. Relevant antibody controls were previously listed in the Methods section (**lines 681-685**) which include human IgG, CDX0239, CDX0239 with deglycosylation via asparagine to alanine mutation, CDX0239 with deglycosylation via enzymatic digestion, and CDX0239 with deglycosylation via asparagine to alanine mutation with conjugated L6 linker.

- The human IgG control was added to demonstrate lack of anti-tumor activity of a non-target full-length human IgG.
- The CDX0239 control was utilized to demonstrate lack of anti-tumor activity of the unconjugated ALK-directed monoclonal antibody.
- CDX0239 with deglycosylation via asparagine to alanine mutation and CDX0239 with deglycosylation via enzymatic digestion were used to demonstrate lack of anti-tumor activity with maximization of antibody-dependent cellular cytotoxicity (ADCC) by enhancing antibody binding to FcγRIIIA (CD16) on leukocytes (**Trastoy et al. Nat Commun. 2023. PMID: 36973249**).
- Lastly, CDX0239 with deglycosylation via asparagine to alanine mutation with linker was included to demonstrate lack of linker-induced anti-tumor activity and confirm conjugation of the PBD dimer payload is required to induce cytotoxicity.

We now include relevance of antibody controls in the Results section (**lines 293-298**) and we have added additional interpretation of the results from our controls in the Discussion section (**lines 429-440**).

Comment 6. *Figure 8 shows the NB-SD model to be sensitive to ADC but resistant to lorlatinib. It would be important to know if the data in 8B and 8D is the same as 5A or an experimental repeat. They look very similar and so if a repeat, this would add to the story and should be made clear in the narrative.*

Response: The data in **Figure 8B** and **Figure 8D** are from the same *in vivo* study presented in **Figure 6A** and **Figure 7A**. The curves are redemonstrated in Figure 8 to highlight the efficacy of CDX0239-PBD in an ALK surface-expressing and ALK-

mutated model that demonstrates *de novo* resistance to small molecular inhibition of ALK, further emphasizing the application of CDX0239-PBD to patients with limited therapeutic options and emphasizing the proof-of-concept that the same antigen that is resistant to small molecular inhibition can be effectively targeted with an ADC regardless of mutational status. We have updated **Figure 8C** and **Figure 8D** with modification of the x-axis scale to weeks to remain consistent with survival curves throughout the manuscript.

REVIEWER 2:

The manuscript is well-structured, and the experiments are generally well designed and executed. However, there are a few aspects that require further clarification and additional data to strengthen the conclusions, particularly regarding the therapeutic potential of CDX0239-PBD in CNS malignancies.

Comment 1. *IC50 Determination: The authors should provide the IC50 value of CDX0239-PBD to allow for a better understanding of its potency.*

Response: IC₅₀ values were previously reported in the Results section (**lines 254-264**) and presented as a table for each cell line (**Supplemental Table 6**).

Comment 2. *Blood-Brain Barrier Considerations: Given that CDX0239-PBD would need to cross the blood-brain barrier to be effective against CNS malignancies, the expected concentration of the drug in the brain should be discussed. Experimental data or relevant pharmacokinetic modeling would strengthen this aspect.*

Response: Neuroblastoma is a disease of the peripheral nervous system, not the central nervous system (CNS). While neuroblastoma can metastasize to the brain, this is rare at diagnosis. With respect to primary CNS malignancies, there is precedent for ADCs having activity against CNS metastases. Trastuzumab deruxtecan is a HER2-directed full-length IgG-based ADC that has been studied in HER2-positive breast cancer with brain metastases and a single arm, Phase II clinical trial demonstrated complete and partial intracranial responses when administered intravenously at standard systemic dosing and is currently being studied in patient population with HER2 low expressing breast cancer with brain metastases (**Bartsch et al. Nature Medicine. 2022. PMID: 35941372**). Additionally, when thinking about CNS malignancies that could benefit from an ALK-directed ADC, we know that selective disruption of the blood-brain barrier (BBB) in some tumors with CNS metastases has been previously established (**Tiwarly et al. Scientific Reports. 2018. PMID: 29844613**). Altogether, these studies suggest that CDX0239-PBD may have enhanced permeability into the CNS in the setting of brain metastases or primary CNS tumors as compared to a BBB unaffected by malignancy or systemic disease. We have now touched on this in the Discussion section (**lines 390-396**). Future studies beyond the scope of this work will evaluate CNS pharmacokinetics in relevant models.

Comment 3. *ALK Expression Assessment: The authors use ALK-D5F3 for immunohistochemical staining in FFPE TMA slides. The staining in Figure 1 appears to be strong, with a moderately high H-score. How does this compare to ALK-amplified or overexpressed tumors? Providing a comparative analysis would enhance the interpretation of these results.*

Response: To address this reviewer's questions and similar comments made by other reviewers, we have now comprehensively annotated our neuroblastoma tumor tissue microarray (TMA) which includes 55 tumors representative of patients with this disease: seven tumors have an underlying *ALK* mutation, and all remaining tumors have wild-type *ALK* with a range of protein expression. The clinical, histopathologic, and molecular features of the tumors embedded in this TMA are representative of the neuroblastoma patient population including diversity in *MYCN* amplification status, *ALK* status, sex, age group, differentiation, histology, ploidy status, mitosis-karyorrhexis index, timing of biopsy (diagnostic, post-chemotherapy, and at relapse), Neuroblastoma Risk Group Staging System (INRGSS) risk stratification, and International Neuroblastoma Risk Group (INRG) stage. None of these features demonstrate statistically significant differences in average H-scores for ALK staining, supporting the conclusion that most neuroblastoma tumors express ALK. We have updated Figure 2 and our results section to include a comprehensive annotation of the TMA H-score analysis by subgroups (**lines 175-188 and Figure 2B**). Additionally, we have updated Figure 2 to include a high-powered pictograph (20x magnification) of a diagnostic neuroblastoma tumor core with an H-score of 142.33 which represents the near average H-score of 144.71 across the TMA (**lines 188-189, Figure 2C**).

The reviewer specifically asks how TMA staining for ALK compares to expression in *ALK*-amplified tumors. Notably, just as *ALK* amplification is rare in patients (~3% of patients with high-risk neuroblastoma, **Berko et al. J Clin Oncol. 2025. PMID: 40036726**), none of the tumors on this TMA harbor *ALK*-amplification. The tumors represent a mixture of *ALK* wild-type and *ALK*-mutant tumors and demonstrate the spectrum of ALK protein expression. *ALK* amplification in human neuroblastoma tumors denotes an increased in the number of copies of the *ALK* gene (gene dosage), typically ranging from 20-100+, which consequently results in dramatic overexpression at the protein level as demonstrated with the NB-1 cell line, one of the only existing models of *ALK* amplification. We previously quantified cell surface ALK expression in NB-1 and a panel of other human neuroblastoma-derived cell lines using quantitative flow cytometry with a chimeric anti-ALK antibody (**Sano et al. Sci Transl Med. 2019. PMID: 30867324**). As expected, NB-1 demonstrated significantly higher ALK cell surface expression when compared with all other ALK wild-type cell lines. This suggests that *ALK*-amplified tumors would likely have H-scores closer to the maximum of 300, and that the average H-score of 144.71 in our patient tumor TMA is representative of ALK overexpression across the majority of neuroblastoma tumors. We clarify the nuances of ALK expression (wild-type versus amplification) in the Discussion section (**line 402-408**).

Comment 4. *Expression Threshold for Efficacy: At what expression level (e.g., copy number threshold) does CDX0239-PBD lose efficacy? This information is critical for understanding patient selection criteria.*

Response: We agree with the reviewer that determining the ALK expression threshold for efficacy is of great importance to evaluate the translational potential of CDX0239-PBD and to serve as a potential biomarker for patient selection. Important to note that “expression level” is directly linked to receptor density on the cell surface. A very small subset of tumors, as noted above, will have true copy number amplification. We explain in detail in response to *Reviewer 1, Comment 4*, why we were unable to conjugate a fluorophore to CDX0239 for quantitative flow cytometric analysis of receptor density (antibodies bound per cell).

Our existing flow cytometry data enabled us to calculate the ALK:IgG geometric mean of ALK surface expression in comparison to non-specific binding of the IgG control in each xenograft model resulting in an effective quantitative representation of ALK surface expression for each xenograft model. We show that CDX0239-PBD leads to complete maintained responses in the NGP *ALK* wild-type (WT) xenograft model (ALK:IgG geometric mean of 8.6), and as expected did not demonstrate activity in the NGP *ALK* knockout (KO) model (ALK:IgG geometric mean of 1.6). CDX0239-PBD has decreased efficacy in the SK-N-AS xenograft (ALK:IgG geometric mean of 3.5). Thus, we can conclude that the therapeutic threshold of ALK surface expression for CDX0239-PBD efficacy lies between an ALK:IgG geometric mean of 3.6-8.6. While immunohistochemistry (IHC) is a more practical method for assessing ALK expression in the clinical setting (as we will not have access to fresh tumors for flow cytometry), it is fraught with many challenges. The extensive data we have for ALK staining in both a neuroblastoma patient-derived xenograft (PDX) tissue microarray (TMA, **Sano et al. Sci Transl Med. 2019. PMID: 30867324**) and a human neuroblastoma tumor TMA in this body of work suggests that most patients with neuroblastoma could benefit from an ALK-directed ADC and that the clinical trial will teach us more precisely about the expression threshold and response. We elaborate on expression threshold and efficacy in the Discussion section (**lines 398-410**).

Comment 5. *Drug-Antibody Ratio (DAR) and Pharmacokinetics: The manuscript states a DAR of 6 for CDX0239-PBD. The authors should provide the in vivo pharmacokinetic (PK) profile, including measurements of free PBD levels, to evaluate potential systemic toxicity and drug clearance.*

Response: PBD is a highly potent DNA crosslinking agent that would induce systemic and end-organ toxicity when in circulation. While we did not measure free PBD levels, we know that CDX0239 has cross species reactivity with human and murine ALK (**Supplemental Figure 1**), allowing us to surveil for on and off-target toxicity. We did not observe significant decreases in mouse weights or signs of toxicity during or after treatment with CDX0239-PBD in any of the xenograft models throughout the study (**lines 308-309 and lines 326-327, Supplemental Figure 6**). These data support stability of the linker and lack of systemic free PBD. When thinking about pharmacokinetic (PK) studies, the two analytes of interest for ADCs are total antibody and conjugate. The development of these analytes is beyond the scope of this work, particularly since the pivotal PK studies needed for Investigational New Drug (IND)-enabling studies will have to be performed and measured in cynomolgus monkeys. We have further elaborated about the importance of the DAR, PK, and measurements of free drug in the Discussion section (**lines 485-488**).

Comment 6. *In Vivo Study Design: The efficacy argument for CDX0239-PBD could be further strengthened by including an additional treatment arm with PBD alone at equimolar concentrations. This would help distinguish the contribution of the ADC from that of the free payload.*

Response: PBD is a potent DNA crosslinking agent with significantly higher potency than other DNA-binding agents that are administered systemically such as cyclophosphamide and cisplatin, and PBD has been precluded from systemic administration due to the significant cardiotoxicity profile in animal models due to hydroxylation at C-9 (Li et al. **Appl Environ Microbiol.** 2009. PMID: 19270142). Therefore, free PBD cannot be given safely, nor at relevant doses, via systemic administration in murine models as it would include significant cardiotoxicity in addition to end-organ toxicity and myelosuppression. Notably, CDX0239 has cross-species reactivity with human and murine ALK (**Supplemental Figure 1**). To distinguish the contribution of the ADC from that of any off-target activity, we **i)** demonstrate lack of cytotoxicity in co-culture with the NGP ALK knockout (KO) cell lines (**Figure 5C**), **ii)** performed an *in vitro* competition study in the ALK-amplified NB-1 cell line demonstrating rescue of neuroblastoma cell viability when unconjugated CDX0239 is added to co-culture with CDX0239-PBD for 5 days (**Figure 5E**), and **iii)** demonstrate complete and sustained tumor regressions and 100% survival in our ALK-expressing xenograft models treated with CDX0239-PBD without significant weight loss or signs of toxicity (**Figures 6-8, Supplemental Figure 6**). These data support linker stability without off-target cleavage and subsequent off-target toxicity.

Comment 7. *Typographical Error: Line 332 contains a minor typo where “am” should be corrected to “an.”*

Response: Corrected.

REVIEWER 3:

In this manuscript, A. Guerra et al. describe a preclinical evaluation of CDX0239-PBD, a novel antibody-drug conjugate (ADC) targeting Anaplastic Lymphoma Kinase (ALK), a lineage-restricted antigen overexpressed in neuroblastoma (NB) and several other pediatric and adult malignancies. Utilizing a humanized ALK-specific monoclonal antibody conjugated to a pyrrolobenzodiazepine (PBD) payload, the study demonstrates potent and selective cytotoxicity in ALK-expressing tumor cells, with minimal off-target effects in normal tissues. The authors provide extensive in vitro and in vivo evidence of ALK surface expression-dependent internalization, lysosomal trafficking, and DNA-damage-mediated apoptosis, leading to sustained tumor regression in multiple xenograft models. These findings suggest that CDX0239-PBD is a promising candidate as a precision therapeutic for ALK-expressing cancers. Overall, the study represents a significant advance in targeted therapies for NB and other ALK-positive malignancies. The experimental design is rigorous, with appropriate positive and negative controls, in vitro and in vivo validation, and the use of multiple ALK-expressing cancer models. Notably, demonstrated minimal ALK expression in normal tissues decreases the risk of CDX0239-PBD on-target toxicity. Mechanistically, the results demonstrate that the ADC works via ALK internalization and lysosomal localization leading to induction of apoptosis in cancer cells. However, there are the following concerns:

Comment 1. *The reliance on transplantable xenograft models limits the understanding of CDX0239-PBD efficacy against spontaneous tumors with their complex TME. Authors could try a syngeneic NB model or at least acknowledge the limitations of xenograft models in discussion.*

Response: We acknowledge the limitations of understanding the efficacy of CDX0239-PBD using subcutaneously implanted xenograft models as compared to spontaneous tumors such as a TH-MYCN transgenic murine model. As suggested, we have added discussion on future utilization of spontaneous tumor models to better recapitulate the complex TME to the Discussion section (**lines 480-483**) and plan these studies as part of our IND enablement of the clinical candidate we will create.

Comment 2. *Figure 2 lacks an analysis of ALK surface expression's association with MYCN amplification and patient age, independent prognostic markers in NB.*

Response: We have updated **Figure 2** and the results section to now include a detailed annotation and analysis of ALK staining H-scores in our neuroblastoma tumor tissue microarray (TMA) by MYCN amplification status, *ALK* status (wild-type and mutated), sex, age group, differentiation, histology, ploidy status, mitosis-karyorrhexis index (MKI), time of biopsy (diagnostic, post-chemotherapy, and at relapse), revised International Neuroblastoma Risk Group Staging System (INRGSS) risk stratification, and International Neuroblastoma Risk Group (INRG) stage in comparison to ganglioneuroblastoma tumors and healthy tissues (placenta, adrenal cortex, adrenal medulla, and tonsil, **lines 175-188, Figure 2B**). Our analysis demonstrates significant ALK protein expression across all subsets (average H-score of 144.71 among 55 neuroblastoma tumors) with no significant differences in ALK staining H-score when compared between genomic, tumor biology, demographic, and diagnostic subgroups.

Comment 3. *Although the study discusses ADCs' ability to overcome TME-related barriers, more in-depth exploration of how CDX0239-PBD penetrates tumor vs. normal tissues (e.g. liver) would strengthen the translational potential.*

Response: We provide a more in-depth explanation of how CDX0239-PBD might overcome TME-related barriers including the extracellular matrix (ECM) and the highly immunosuppressive environment in neuroblastoma in the Discussion section (**lines 380-390**). ADCs can overcome these barriers by enabling bystander cytotoxicity to tumor cells with low surface target antigen (Li et al. **Cancer Res.** 2016. PMID: 26921341) in addition to tumor cell death with subsequent release of damage-associated molecular patterns (DAMPs) to stimulate dendritic cell activation that facilitate antigen uptake and migration to the lymph nodes, ADC-mediated reduction of Tregs, augmentation of MHC class I expression to promote activity of cytotoxic T cells, and enhancement of leukocyte infiltration, expansion, and interferon gamma (IFN γ) production (Chang et al. **J Clin Invest.** 2023. PMID: 37712425). Our *in vivo* studies demonstrate potent efficacy of CDX0239-PBD with complete maintained responses in moderate to high surface ALK-expressing xenograft models and demonstrate an initial transient response in low surface ALK-expressing xenograft models suggesting the presence of bystander cytotoxicity that overcomes structural and spatial barriers of the TME in addition to cytotoxicity in tumor cells with low surface target antigen. We will explore additional TME-directed and immune-enhancing properties of our ALK-directed ADC further in future studies with spontaneous tumor and immunocompetent animal models. We address this in depth in the Discussion section (**lines 402-410 and lines 480-483**).

REVIEWER 4:

The submitted manuscript, "A humanized ALK-directed antibody-drug conjugate CDX0239-PDB demonstrates potent efficacy in ALK-expressing cancers" by Guerra and colleagues, presents a compelling approach to precision chemotherapy development. Over more than a decade, the authors have advanced this research toward a potential clinical application. Beginning with a murine antibody capable of blocking ALK activity, the researchers have systematically improved its properties through modifications, enhancing its affinity and specificity. Ultimately, they generated a humanized ALK-directed monoclonal antibody with high affinity, conjugated to pyrrolobenzodiazepine. The manuscript presents promising results, demonstrating significant responses in ALK-expressing cell line xenografts and opening a viable path for clinical development.

Comment 1. *Lines 95–96: It seems odd that reference 2 appears both in the middle and at the end of the sentence. The authors are making a point, but scientists in the field are already aware of these facts.*

Response: We have now removed the citation of reference 2 from the middle of the sentence and retained its citation at the end of the sentence with the additional cited studies which together demonstrate *ALK* as the major cause of hereditary neuroblastoma and as the gene with the most common somatic single nucleotide mutations in the sporadic forms of the disease.

Comment 2. *Lines 104–106: The reference is incorrect. Where has this manuscript been published? The authors should also include additional references, as this observation has been reported elsewhere, such as in PMID: 20719933, PMID: 26121087, and PMID: 25071110.*

Response: The study referenced has now been published in the Journal of Clinical Oncology (**Berko et al. J Clin Oncol. 2025. PMID: 40036726**) and the reference has been updated accordingly. We have added additional references in addition to our own recent publication and highlight that MYCN amplification is more frequent with ALK aberrant tumors, particularly those harboring ALK F1174 mutations. Previous studies have demonstrated higher frequencies in ALK and RAS-MAPK pathway mutations at relapse, suggesting an enrichment of selected subclonal mutations as a mechanism of therapy resistance (**lines 109-112**).

Comment 3. *Line 110: The authors should add another reference to emphasize that most patients harboring neuroblastoma tumors express ALK (PMID: 38229432).*

Response: We have added this reference to this statement to further emphasize that native ALK protein is expressed on the cell surface of most neuroblastoma tumors (**line 116**).

Comment 4. *Lines 120–125: This sentence could be improved for clarity, making it more accessible to general readers. Additionally, several long sentences throughout the manuscript should be refined for readability.*

Response: We have modified this statement and multiple statements throughout the manuscript to improve clarity.

Comment 5. *Line 161: Add reference PMID: 38229432, which documents ALK expression in neuroblastoma patients. This should be included, as the authors validate previously published findings effectively.*

Response: This reference has now been added to this statement (**line 173**)

Comment 6. *Line 184: While the methods section clarifies this to some extent, the authors should provide more details on the methodology and procedure for generating CDX0239. Was CDX0239 derived from the discovery of CDX125, as referenced in reference 31, or from PMID: 30867324? Additionally, where was the murine anti-ALK antibody originally obtained?*

Response: We have expanded the methods section to provide additional information on the origin, methodology, and procedure for generating humanized CDX0239 which was derived from the discovery of the chimeric antibody CDX0125 (**lines 549-568**).

Comment 7. *Line 185: The authors use the bio-layer interferometry method. It would be appropriate to cite a more foundational article describing this technique, such as PMID: 22414486.*

Response: This foundational study is now included in our citations.

Comment 8. *Lines 185–203: Is CDX0239 activating or inhibiting ALK activity?*

Response: Previous studies have demonstrated that the chimeric ALK-directed monoclonal antibody CDX123 is an ‘activating’ antibody that likely functions by crosslinking of receptor monomers without affecting ligand recognition, and the chimeric antibody CDX0125 is an ‘inhibitory’ antibody leading to decreased ALK phosphorylation due to binding overlap at the ALKAL binding site (**Li et al. Nature. 2022. PMID: 34819665; Sano et al. Sci Transl Med. 2019. PMID: 30867324**). The ‘activating’ properties of CDX123 and the ‘inhibitory’ effects of CDX0125 were demonstrated in the

neuroblastoma cell line IMR-32 (*ALK* wild-type) which does not harbor an *ALK* mutations and requires ligand binding for activation. CDX0239 is a humanized antibody derived from the discovery of the chimeric antibody CDX0125. We demonstrate that CDX0239 does not have an effect on phosphorylated *ALK* levels in the *ALK*-mutated (c.3522C>A, F1174L) NB-SD xenograft (**Harenza et al. *Sci Data*. 2017. PMID: 28350380**). Given NB-SD is *ALK*-mutated, there is constitutive *ALK* activation without the presence of a ligand compared to wild-type *ALK*, we would not expect CDX0239 to induce decreased levels of phosphorylated *ALK* even if CDX0239 is bound to the *ALK* binding site. This is suggested by no evidence of decreased phosphorylated *ALK* expression after treatment with CDX0239-PBD (**Figure 8E**). Additionally, we demonstrate no evidence of anti-tumor activity with unconjugated CDX0239 alone in the *ALK*-amplified NB-1 neuroblastoma cell line and matched xenograft model (**Figure 5B**). If CDX0239 had a significant inhibitory effect on phosphorylated *ALK* signaling, there would likely be an observed anti-tumor activity as previously demonstrated with small molecular inhibition of *ALK* in NB-1 (**Infarinato et al. *Cancer Discovery*. 2016. PMID: 26554404**). This is now clarified in the Discussion section (**lines 429-436**). Lastly, we demonstrate that binding of CDX0239 to surface *ALK* leads to endocytosis of the *ALK* receptor-antibody complex with intracellular localization to the lysosome (**Figure 4 and Supplemental Figure 3**) which leads to a reduction in total *ALK* expression (**Figure 8E**).

Comment 9. Line 313: CDX0239-PBD did not affect phosphorylated *ALK* (p*ALK*1604) levels. What about phosphorylated p*ALK*1278, which is necessary for *ALK* activity? The authors should verify their data using a p*ALK*1278 antibody. Have they observed any additional differences in downstream *ALK* signaling when using CDX0239? This is not addressed in the manuscript.

Response: We and others have shown that there is variability in expression of the two key *ALK* phospho-tyrosine sites, namely p*ALK*128 and p*ALK*1604, and this differs by model and underlying activating *ALK* mutation. We attempted numerous times to analyze the processed NB-SD tumor lysates for p*ALK*1278 expression, however we were unable to detect this phospho-site by Western Blot. NB-SD harbors a c.3522C>A, F1174L mutation which confers constitutive activation with aberrant downstream signaling including the PI3K/AKT, RAS/MAPK, and JAK/STAT pathways (**Harenza et al. *Sci Data*. 2017. PMID: 28350380; Debruyne et al. *Oncogene*. 2015. PMID: 26616860**). To demonstrate any potential effect of CDX0239 on downstream *ALK* signaling, we analyzed the NB-SD tumor lysate samples for phosphorylated BRAF (p445, Cell Signaling, #2696S) as BRAF is downstream of *ALK* in the RAS/MAPK signaling pathway. We found decreased expression of pBRAF 7 days after treatment with CDX0239-PBD (Figure included below) which is consistent with our previous results demonstrating diminished total *ALK* expression 7 days after treatment (**Figure 8E**). Given that there was no effect on p*ALK*1604 expression after treatment with CDX0239-PBD, we postulate that the diminished expression of phosphorylated BRAF 7 days after treatment with CDX0239-PBD is due to antibody-receptor internalization which leads to diminished total *ALK* expression and thus diminished signaling of downstream proteins. Further characterization of additional signaling proteins downstream of *ALK* is out of scope for this study, particularly since *ALK* signaling within the context of activating mutations in the full-length receptor remains undefined.

Comment 10. Line 398: The authors should include reference PMID: 34646012, as it was published in the same issue as reference 59.

Response: We have added this reference.

Comment 11. Line 453: The specificity and selection process for the humanized antibody (CDX0239) are unclear. Which

ALK epitope does it bind to? Is it ALK or ALK/ALKAL2? The authors refer to reference 31, where they express a codon-optimized cDNA encoding the extracellular glycine-rich domain of ALK (aa 678-1030) fused to the ALK ligand ALKAL2 (aa 71-152). Could it be that the antibody only recognizes ALK RTK when bound to an ALK ligand? The authors should clarify this.

Response: We have added more details to describe the selection process for the humanized antibody CDX0239. The original ALK-directed murine antibody was generated against human ALK (T637-S1038) comprising the glycine rich and epidermal growth factor domain as stated above (**Sano et al. *Sci Transl Med.* 2019. PMID: 30867324**). Although the crystal structure of the ALK extracellular domain bound to CDX0239 is not available, the chimeric version of the anti-ALK antibody (CDX0125) was shown to protect a peptide in the C-terminal loop of the TNF- α like region of ALK (Y966-V972) which overlaps with the activating peptide (ALKAL) binding site in hydrogen/deuterium exchange mass spectrometry (HDX-MS; **Li et al. *Nature.* 2021. PMID: 34819665; Figure 4**), indicating that the antibody binds to the extracellular domain proximal to the ALK cleavage site (N654-L655; **Huang et al. *Cell Rep.* 2021. PMID: 34260934**). Our studies demonstrate that CDX0239 binds to the receptor tyrosine kinase (RTK) in the absence of an ALK ligand as we successfully utilized CDX0239 to bind surface ALK to determine surface ALK expression via flow cytometry (**Figure 3**). We have further clarified this in the manuscript in the Methods section (**lines 549-561**).